# A New Model of Interval-Valued Intuitionistic Fuzzy Weighted Operators and Their Application in Dynamic Fusion Target Threat Assessment

**DOI:** 10.3390/e24121825

**Published:** 2022-12-14

**Authors:** Chengli Fan, Qiang Fu, Yafei Song, Yingqi Lu, Wei Li, Xiaowen Zhu

**Affiliations:** Air Defense and Antimissile College, Air Force Engineering University, Xi’an 710051, China

**Keywords:** interval-valued intuitionistic fuzzy aggregation operator, threat assessment, multi-time fusion, interval-valued intuitionistic fuzzy entropy, interval-valued intuitionistic fuzzy number distance measurement

## Abstract

Existing missile defense target threat assessment methods ignore the target timing and battlefield changes, leading to low assessment accuracy. In order to overcome this problem, a dynamic multi-time fusion target threat assessment method is proposed. In this method, a new interval valued intuitionistic fuzzy weighted averaging operator is proposed to effectively aggregate multi-source uncertain information; an interval-valued intuitionistic fuzzy entropy based on a cosine function (IVIFECF) is designed to determine the target attribute weight; an improved interval-valued intuitionistic fuzzy number distance measurement model is constructed to improve the discrimination of assessment results. Specifically, first of all, we define new interval-valued intuitionistic fuzzy operation rules based on algebraic operations. We use these rules to provide a new model of interval-valued intuitionistic fuzzy weighted arithmetic averaging (IVIFWAA) and geometric averaging (IVIFWGA) operators, and prove a number of algebraic properties of these operators. Then, considering the subjective and objective weights of the incoming target, a comprehensive weight model of target attributes based on IVIFECF is proposed, and the Poisson distribution method is used to solve the time series weights to process multi-time situation information. On this basis, the IVIFWAA and IVIFWGA operators are used to aggregate the decision information from multiple times and multiple decision makers. Finally, based on the improved TOPSIS method, the interval-valued intuitionistic fuzzy numbers are ordered, and the weighted multi-time fusion target threat assessment result is obtained. Simulation results of comparison show that the proposed method can effectively improve the reliability and accuracy of target threat assessment in missile defense.

## 1. Introduction

Target threat assessment is the third level of the JDL information fusion model, and is a core issue of missile defense command decision-making. Methods to quickly and accurately evaluate the threat level of an incoming target represent a difficult technical problem that restricts the improvement of the combat effectiveness of missile defense weapon systems. Due to numerous characteristic attributes of ballistic targets and missile defense warfare being a continuous and dynamic process, the threat assessment of a target requires comprehensive consideration of multiple factors, which can be formulated as a type of uncertain dynamic multi-attribute group decision-making problem [1,2].

Commonly used threat assessment methods include Bayesian networks [3], rough set theory [4], D-S evidence theory [5], fuzzy reasoning [6], and multi-attribute decision-making [7]. Although the Bayesian networks can effectively handle uncertain information, the selection of transition probability depends on expert experience, the reliability is poor and it is only applicable to situations where there is a large amount of sample data, and the solution accuracy for small sample data is poor. Thus, it is not suitable for missile defense warfare with a lack of warfare sample data. Although rough set theory does not require prior information outside the data set, it needs to build a large knowledge base to support the construction of relevant rules, and it’s difficult to adapt to the high requirements of missile defense warfare for decision-making timeliness. The D-S evidence theory is weak in processing conflicting information, and when the problem space is large the problem of the combinatorial explosion will occur. Considering the large threat of ballistic missiles, one missed interception will cause great damage to surface defense assets and require high threat assessment accuracy. Therefore, a variety of factors need to be considered. Fuzzy set theory combined with dynamic multi-attribute group decision theory can be used to solve this problem.

Fuzzy sets have been widely used to describe and process fuzzy uncertain decision information since Zadeh [8] proposed it. Atanassov [9] extended the fuzzy set to the intuitionistic fuzzy set by removing the constraint condition that the sum of membership and non-membership is one. Intuitionistic fuzzy sets can describe the uncertainty of decision-making information in more detail, so they have been widely used in intelligence reasoning, decision-making, and other fields. In recent years, a large number of multi-attribute group decision-making methods based on intuitionistic fuzzy sets have been proposed. Pamucar [10] provided multi-criteria decision-making that combines interval grey numbers and normalized weighted geometric Dombi-Bonferroni mean operator. Shen et al. [11] proposed an extended intuitionistic fuzzy TOPSIS method. In et al. builds an intuitionistic fuzzy multi-attribute group decision-making model based on the consistency method. Jin et al. [12] proposed an intuitionistic fuzzy preference relation group decision-making method based on multiplicative consistency. Wu et al. [13] used a multi-criteria decision-making model of triangular intuitionistic fuzzy number correlation. Based on this, Kong et al. [14] proposed a quantification method of multi-attribute threat indicators that addresses the issue of threat assessment indicators of ground combat targets being diverse and difficult to quantify, and unified the quantitative results of indicators in the form of intuitionistic fuzzy sets. Wang et al. [15] considered the preferences of decision makers and studied threat assessment methods with unknown target attribute weights.

In order to overcome the problem of threat assessment methods relying too much on expert knowledge, Xiao et al. [16] proposed an air target threat assessment method based on intuitionistic fuzzy hierarchical analytic processes. In [17], Zhang et al. used intuitionistic fuzzy entropy to calculate attribute weights and constructed a target threat assessment model. In the context of intuitionistic fuzzy multi-attribute decision-making, [18] proposed a target threat assessment method based on tripartite decision-making.

The above methods are all useful attempts to combine fuzzy set theory with multi-attribute decision-making theory. However, for missile defense warfare with dynamic, time-sensitive, and strong antagonism, the existing threat assessment methods still have a number of shortcomings. First, due to the missile defense combat environment, the high complexity, and the limitations of sensor detection performance, the battlefield information present is incomplete and uncertain. The multi-attribute group decision-making method based on intuitionistic fuzzy sets uses a certain “point value” to represent the assessment data, which will lead to the excessive deviation between the assessment result and objective reality. Therefore, in the threat assessment of ballistic targets, it is necessary to expand the point value data into interval-valued intuitionistic fuzzy numbers, and to smooth the detection data parameters to control the error within a certain range in order to improve the accuracy and reliability of the threat assessment. Second, although a large number of interval-valued intuitionistic fuzzy operators [19] have been proposed in recent years, and most of them are defined in the traditional interval-valued intuitionistic fuzzy operation rules, they ignore the relationship between aggregated data when performing information fusion. Because of this, there will be results contrary to intuitive analysis and the ambiguous nature of the complex battlefield will not be meticulously and flexibly reflected. Third, most of the literature [1,2,3,4,5,6,7,14,15,16] only focuses on the information in the current moment when conducting the threat assessment, ignoring the timing of target information, making it difficult to obtain objective and comprehensive threat assessment results. Fourth, although the existing interval-valued intuitionistic fuzzy entropy has various forms [20,21,22], when the deviation of the degree of membership and the degree of non-membership is equal there is inconsistency with the intuitive facts.

Based on the above analysis, the contributions of the work in this paper are as follows: (1) Based on a new interval-valued intuitionistic fuzzy operation rules that retain the properties of classical algebra, the new interval-valued intuitionistic fuzzy arithmetic weighted average (IVIFWAA) operator and the interval-valued intuitionistic fuzzy geometric weighted average (IVIFWGA) operator are proposed to perform nonlinear aggregation operation on interval intuitionistic fuzzy numbers. (2) The threat assessment index system for typical missile defense combat elements is constructed, and the interval-valued intuitionistic fuzzy entropy based on cosine function (IVIFECF) is proposed to solve the problem of inconsistency with intuitive facts when the deviation between membership and non-membership are equal. (3) Considering the objective, subjective and time series weights, a comprehensive weight model of target attributes based on IVIFECF is constructed to process multi-time situation information. (4) An ordering method of interval-valued intuitionistic fuzzy numbers based on improved TOPSIS is proposed to improve the discrimination ability between decision-making results. (5) The evaluation model based on IVIFEC-IVIFWA-TOPSIS is constructed by aggregating the multi-target attributes and multi-expert decision-making information multiple times, which improves the reliability and accuracy of missile defense target threat evaluation.

We conclude this section by outlining the remainder of the article. Section 2 introduces the related concepts of interval-valued intuitionistic fuzzy sets(IVIFSs). Section 3 defines new interval-valued intuitionistic fuzzy operation rules. Section 4 defines new interval-valued intuitionistic fuzzy weighted average operators and proves their algebraic properties. In Section 5, we present a missile defense dynamic multi-time fusion target threat assessment method based on IVIFECF-IVIFWA-TOPSIS. Numerical examples and an analysis of the performance of the proposed algorithm are given in Section 6. We summarize the results and provide conclusions in the seventh part.

## 2. Preliminaries

### 2.1. Interval-Valued Intuitionistic Fuzzy Sets

**Definition 1. [23].** Let $X=\{x_{1},x_{2},\cdots ,x_{n}\}$
*be a non-empty set. Given X, the interval-valued intuitionistic fuzzy set*
$\tilde{A}$
*can be represented as*
$$\tilde{A}=\left\{<x,{\tilde{\mu }}_{\tilde{A}}(x),{\tilde{v}}_{\tilde{A}}(x)>\left|x\in X\right.\right\}$$
*where*
${\tilde{\mu }}_{\tilde{A}}(x)\subset [0,1]$
*and*
${\tilde{v}}_{\tilde{A}}(x)\subset [0,1]$
*are the*
$\tilde{A}$
*membership and non-membership intervals, respectively associated with*
x
*in*
X*, Further, for any*
x∈X*, the condition*
$0\leq \sup{\tilde{\mu }}_{\tilde{A}}(x)+\sup{\tilde{v}}_{\tilde{A}}(x)\leq 1$
*is satisfied. For ease of explanation, the interval-valued intuitionistic fuzzy set can be written as:*$$\tilde{A}=\left\{<x,[\mu_{{}_{\tilde{A}}}^{L}(x),\mu_{{}_{\tilde{A}}}^{U}(x)],[v_{{}_{\tilde{A}}}^{L}(x),v_{{}_{\tilde{A}}}^{U}(x)]>\left|x\in X\right.\right\}$$
*where*
$\mu_{{}_{\tilde{A}}}^{U}(x)+v_{{}_{\tilde{A}}}^{U}(x)\leq 1,\mu_{{}_{\tilde{A}}}^{L}(x)\geq 0,v_{{}_{\tilde{A}}}^{L}(x)\geq 0$.

The intuitionistic fuzzy interval can be expressed as
$${\tilde{\pi }}_{\tilde{A}}(x)=1-{\tilde{\mu }}_{\tilde{A}}(x)-{\tilde{v}}_{\tilde{A}}(x)=[1-\mu_{{}_{\tilde{A}}}^{U}(x)-v_{{}_{\tilde{A}}}^{U}(x),1-\mu_{{}_{\tilde{A}}}^{L}(x)-v_{{}_{\tilde{A}}}^{L}(x)]$$

Generally, the ordered pair $\left([\mu_{{}_{\tilde{A}}}^{L}(x),\mu_{{}_{\tilde{A}}}^{U}(x)],[v_{{}_{\tilde{A}}}^{L}(x),v_{{}_{\tilde{A}}}^{U}(x)]\right)$ formed by the degree of membership interval $\left[\mu_{{}_{\tilde{A}}}^{L}(x),\mu_{{}_{\tilde{A}}}^{U}(x)\right]$ and non-membership interval $\left[v_{{}_{\tilde{A}}}^{L}(x),v_{{}_{\tilde{A}}}^{U}(x)\right]$ for x
*in* X of $\tilde{A}$ is called an interval-valued intuitionistic fuzzy number. For convenience, let *IVIFS(X)* be the interval-valued intuitionistic fuzzy set on the domain, X; ([a,b],[c,d]) is the interval-valued intuitionistic fuzzy number, where [a,b]⊂[0,1], [c,d]⊂[0,1] and b+d≤1.

**Definition 2. [24].** Let $\tilde{A},\;\tilde{B}\in IVIFS(X)$*, then*
(1)$\tilde{A}\subseteq \;\tilde{B}$ *if and only if *$\mu_{{}_{\tilde{A}}}^{L}(x)\leq \mu_{{}_{\tilde{B}}}^{L}(x)$*, *$\mu_{{}_{\tilde{A}}}^{U}(x)\leq \mu_{{}_{\tilde{B}}}^{U}(x)$*, *$v_{{}_{\tilde{A}}}^{L}(x)\geq v_{{}_{\tilde{B}}}^{L}(x)$*, *$v_{{}_{\tilde{A}}}^{U}(x)\geq v_{{}_{\tilde{B}}}^{U}(x),\forall x\in X$(2)$\tilde{A}=\;\tilde{B}$* if and only if *$\tilde{A}\subseteq \;\tilde{B}$*and, *$\tilde{B}\subseteq \;\tilde{A}$*, *∀x∈X(3)${\tilde{A}}^{C}=\{<x,[v_{{}_{\tilde{A}}}^{L}(x),v_{{}_{\tilde{A}}}^{U}(x)],[\mu_{{}_{\tilde{A}}}^{L}(x),\mu_{{}_{\tilde{A}}}^{U}(x)]>\left|x\in X\right.\}$


**Definition 3. [24].** *Let*${\tilde{\alpha }}_{i}=([a_{i},b_{i}],[c_{i},d_{i}])\;\;(i=1,2,\cdots n)$* be a set of interval-valued intuitionistic fuzzy numbers,* $\omega_{i}$* be the weight of *${\tilde{\alpha }}_{i}$*, with *$\omega_{i}\in [0,1]$*and *$\sum_{i=1}^{n}\omega_{i}=1$*. Then, the interval-valued intuitionistic fuzzy weighted average operator IFWAA: *${\tilde{\Theta }}^{n}\to \tilde{\Theta }$*and the interval-valued intuitionistic fuzzy weighted geometric operator IIFWG:*${\tilde{\Theta }}^{n}\to \tilde{\Theta }$*are respectively: *$$\mathrm{IIFWA}({\tilde{\alpha }}_{1},{\tilde{\alpha }}_{2},\cdots ,{\tilde{\alpha }}_{n})=\left(\left[1-\prod_{i=1}^{n}{\left(1-a_{i}\right)}^{\omega_{i}},1-\prod_{i=1}^{n}{\left(1-b_{i}\right)}^{\omega_{i}}\right]\right.\;,\;\;\;\left.\left[\prod_{i=1}^{n}c_{i}{}^{\omega_{i}},\prod_{i=1}^{n}d_{i}{}^{\omega_{i}}\right]\right)$$$$IIFWG({\tilde{\alpha }}_{1},{\tilde{\alpha }}_{2},\cdots ,{\tilde{\alpha }}_{n})=\left(\left[\prod_{i=1}^{n}a_{i}{}^{\omega_{i}},\prod_{i=1}^{n}b_{i}{}^{\omega_{i}}\right]\right.,\;\;\;\;\;\left.\left[1-\prod_{i=1}^{n}{\left(1-c_{i}\right)}^{\omega_{i}},1-\prod_{i=1}^{n}{\left(1-d_{i}\right)}^{\omega_{i}}\right]\right)$$

### 2.2. The Order of Interval-Valued Intuitionistic Fuzzy Numbers

The aggregated result of interval-valued intuitionistic fuzzy information is still an interval-valued intuitionistic fuzzy number, so the ordering of interval-valued intuitionistic fuzzy numbers is of great significance to fuzzy decision-making. The score function and the exact function are classic methods to realize the ordering of interval-valued intuitionistic fuzzy numbers.

**Definition 4. [24].** *Let*${\tilde{\alpha }}_{1}=([a_{1},b_{1}],[c_{1},d_{1}])$* and *${\tilde{\alpha }}_{2}=([a_{2},b_{2}],$ $\left[c_{2},d_{2}]\right)$* be any two interval-valued intuitionistic fuzzy numbers. The score functions of *${\tilde{\alpha }}_{1}$*, *${\tilde{\alpha }}_{2}$* are expressed as *$s({\tilde{\alpha }}_{1})=\left(a_{1}-c_{1}+b_{1}-d_{1}\right)/2$* and *$s({\tilde{\alpha }}_{2})=$ $\left(a_{2}-c_{2}+b_{2}-d_{2}\right)/2$*, respectively; the exact functions of *${\tilde{\alpha }}_{1}$*, *${\tilde{\alpha }}_{2}$* are expressed as *$h({\tilde{\alpha }}_{1})$ $h({\tilde{\alpha }}_{1})=\left(a_{1}+c_{1}+b_{1}+d_{1}\right)/2$* and *$h({\tilde{\alpha }}_{2})=\left(a_{2}+c_{2}+b_{2}+d_{2}\right)/2$*, respectively, then: *(1)*If *$s({\tilde{\alpha }}_{1})<s({\tilde{\alpha }}_{2})$*, then *${\tilde{\alpha }}_{1}<{\tilde{\alpha }}_{2}$*;*(2)*If* $s({\tilde{\alpha }}_{1})=s({\tilde{\alpha }}_{2})$*, then* ① *If*
$h({\tilde{\alpha }}_{1})<h({\tilde{\alpha }}_{2})$
*, then *
${\tilde{\alpha }}_{1}<{\tilde{\alpha }}_{2}$
*;* ② *If*
$h({\tilde{\alpha }}_{1})=h({\tilde{\alpha }}_{2})$
*, then *
${\tilde{\alpha }}_{1}∼{\tilde{\alpha }}_{2}$.

## 3. New Operations of Interval-Valued Intuitionistic Fuzzy Numbers

**Definition 5. [23].** 
*Let*

$\tilde{\alpha }=([a,b],[c,d])$

*, *

${\tilde{\alpha }}_{1}=([a_{1},b_{1}],[c_{1},d_{1}])$

* and *

${\tilde{\alpha }}_{2}=([a_{2},b_{2}],[c_{2},d_{2}])$

* be interval-valued intuitionistic fuzzy numbers, then *
*(1)* 

${\tilde{\alpha }}_{1}+{\tilde{\alpha }}_{2}=([a_{1}+a_{2}-a_{1}a_{2},b_{1}+b_{2}-b_{1}b_{2}],[c_{1}c_{2},d_{1}d_{2}])$

*(2)* 

${\tilde{\alpha }}_{1}{\tilde{\alpha }}_{2}=([a_{1}a_{2},b_{1}b_{2}],[c_{1}+c_{2}-c_{1}c_{2},d_{1}+d_{2}-d_{1}d_{2}])$

*(3)* 

$\lambda \tilde{\alpha }=([1-{\left(1-a\right)}^{\lambda },1-{\left(1-b\right)}^{\lambda }],[c^{\lambda },d^{\lambda }])$

*(4)* 

${\tilde{\alpha }}^{\lambda }=([a^{\lambda },b^{\lambda }],[1-{\left(1-c\right)}^{\lambda },1-{\left(1-d\right)}^{\lambda }])$




Definition 5 gives the basic rules of interval-valued intuitionistic fuzzy operations. Based on these operating rules, a large number of interval-valued intuitionistic fuzzy aggregation operations can be defined. It should be noted though that the rules for aggregation of interval-valued intuitionistic fuzzy information are not unique. At present, the operation rules based on the Einstein t-norm and the interactive operation rules based on the degree of membership and non-membership have been successively proposed [25,26,27] and used to solve the problem of interval-valued intuitionistic fuzzy multi-attribute decision-making. However, the relationship between new operations and classic operations is not clear. Some interactive operations lack the algebraic properties of classic operations, which has an impact on decision analysis. Therefore, this paper proposes a new interval-valued intuitionistic fuzzy operation rule based on algebraic operations and analyzes its operational characteristics in depth.

**Definition 6.** 
*Let*

${\tilde{\alpha }}_{1}=([a_{1},b_{1}],[c_{1},d_{1}])$

*and*

${\tilde{\alpha }}_{2}=([a_{2},b_{2}],[c_{2},d_{2}])$

*be any two interval-valued intuitionistic fuzzy numbers and respectively define the addition operation “ *

⊕

*” and multiplication operation “*

⊗

*” as *

$${\tilde{\alpha }}_{1}⊕{\tilde{\alpha }}_{2}=\left(\left[\frac{a_{1}+a_{2}-2a_{1}a_{2}}{1-a_{1}a_{2}},\frac{b_{1}+b_{2}-2b_{1}b_{2}}{1-b_{1}b_{2}}\right]\right.,\;\;\left.\left[\frac{c_{1}c_{2}}{c_{1}+c_{2}-c_{1}c_{2}},\frac{d_{1}d_{2}}{d_{1}+d_{2}-d_{1}d_{2}}\right]\right)$$


$${\tilde{\alpha }}_{1}⊗{\tilde{\alpha }}_{2}=\left(\left[\frac{a_{1}a_{2}}{a_{1}+a_{2}-a_{1}a_{2}},\frac{b_{1}b_{2}}{b_{1}+b_{2}-b_{1}b_{2}}\right]\right.,\;\;\;\;\;\left.\left[\frac{c_{1}+c_{2}-2c_{1}c_{2}}{1-c_{1}c_{2}},\frac{d_{1}+d_{2}-2d_{1}d_{2}}{1-d_{1}d_{2}}\right]\right)$$



**Explanation 1.** For functions $f(x,y)=\frac{x+y-2xy}{1-xy}$ and $g(x,y)=\frac{xy}{x+y-xy}$, we can obtain
$$\left\{\begin{array}{l} g(x,y)=1-f(1-x,1-y)\\ f(x,y)=1-g(1-x,1-y) \end{array}\right.$$Therefore, Definition 6 can be expressed as
$${\tilde{\alpha }}_{1}⊕{\tilde{\alpha }}_{2}=\left(\left[f\left(a_{1},a_{2}\right),f\left(b_{1},b_{2}\right)\right],\right.\;\;\;\left.\left[1-f\left(1-c_{1},1-c_{2}\right),1-f\left(1-d_{1},1-d_{2}\right)\right]\right)$$
$${\tilde{\alpha }}_{1}⊗{\tilde{\alpha }}_{2}=\left(\left[g\left(a_{1},a_{2}\right),g\left(b_{1},b_{2}\right)\right],\right.\;\left.\left[1-g\left(1-c_{1},1-c_{2}\right),1-g\left(1-d_{1},1-d_{2}\right)\right]\right)$$

**Explanation 2.** The behavior of the functions $f(x,y)=\frac{x+y-2xy}{1-xy}$ and $g(x,y)=\frac{xy}{x+y-xy}$ where x=y=1 and x=y=0, respectively are not significant. Because $\underset{\left(x,y)\to (1,1\right)}{\lim}f(x,y)=1$, $\underset{\left(x,y)\to (0,0\right)}{\lim}g(x,y)=0$, we can define
$$\left\{\begin{array}{l} \left(\left[0,0\right],\left[1,1\right]\right)⊕\left(\left[0,0\right],\left[1,1\right]\right)=\left(\left[0,0\right],\left[1,1\right]\right)\\ \left(\left[1,1\right],\left[0,0\right]\right)⊕\left(\left[1,1\right],\left[0,0\right]\right)=\left(\left[1,1\right],\left[0,0\right]\right) \end{array}\right.$$

**Theorem 1.** *Given the interval-valued intuitionistic fuzzy numbers *${\tilde{\alpha }}_{1}=([a_{1},b_{1}],$ $\left[c_{1},d_{1}]\right)$* and *${\tilde{\alpha }}_{2}=([a_{2},b_{2}],[c_{2},d_{2}])$*, we can obtain *$${\tilde{\alpha }}_{1}⊗{\tilde{\alpha }}_{2}={\left({\tilde{\alpha }}_{1}{}^{C}⊕{\tilde{\alpha }}_{2}{}^{C}\right)}^{C},{\tilde{\alpha }}_{1}⊕{\tilde{\alpha }}_{2}={\left({\tilde{\alpha }}_{1}{}^{C}⊗{\tilde{\alpha }}_{2}{}^{C}\right)}^{C}$$*where*${\tilde{\alpha }}_{1}{}^{C}=([c_{1},d_{1}],[a_{1},b_{1}])$, ${\tilde{\alpha }}_{2}{}^{C}=([c_{2},d_{2}],[a_{2},b_{2}])$.

**Proof of Theorem 1:** By Definition 6 we obtain: $${\tilde{\alpha }}_{1}{}^{C}⊕{\tilde{\alpha }}_{2}{}^{C}=\left(\left[\frac{c_{1}+c_{2}-2c_{1}c_{2}}{1-c_{1}c_{2}},\frac{d_{1}+d_{2}-2d_{1}d_{2}}{1-d_{1}d_{2}}\right]\right.,\;\left.\left[\frac{a_{1}a_{2}}{a_{1}+a_{2}-a_{1}a_{2}},\frac{b_{1}b_{2}}{b_{1}+b_{2}-b_{1}b_{2}}\right]\right)$$Then, it follows that:$${\left({\tilde{\alpha }}_{1}{}^{C}⊕{\tilde{\alpha }}_{2}{}^{C}\right)}^{C}=\left(\left[\frac{a_{1}a_{2}}{a_{1}+a_{2}-a_{1}a_{2}},\frac{b_{1}b_{2}}{b_{1}+b_{2}-b_{1}b_{2}}\right]\right.,\left.\left[\frac{c_{1}+c_{2}-2c_{1}c_{2}}{1-c_{1}c_{2}},\frac{d_{1}+d_{2}-2d_{1}d_{2}}{1-d_{1}d_{2}}\right]\right)={\tilde{\alpha }}_{1}⊗{\tilde{\alpha }}_{2}$$Therefore: ${\tilde{\alpha }}_{1}⊗{\tilde{\alpha }}_{2}={\left({\tilde{\alpha }}_{1}{}^{C}⊕{\tilde{\alpha }}_{2}{}^{C}\right)}^{C}$.By the same logic: ${\tilde{\alpha }}_{1}⊕{\tilde{\alpha }}_{2}={\left({\tilde{\alpha }}_{1}{}^{C}⊗{\tilde{\alpha }}_{2}{}^{C}\right)}^{C}$, which completes the proof. □

**Theorem 2.** 
*For the interval-valued intuitionistic fuzzy number*

$\tilde{\alpha }=([a,b],[c,d])$

*, we can obtain*

$$n⊙\tilde{\alpha }=\underset{n}{\underset{︸}{\tilde{\alpha }⊕\tilde{\alpha }⊕\cdots ⊕\tilde{\alpha }}}=\left(\left[\frac{na}{1+(n-1)a},\frac{nb}{1+(n-1)b}\right]\right.,\;\;\left.\left[\frac{c}{n-\left(n-1\right)c},\frac{d}{n-\left(n-1\right)d}\right]\right)$$



**Proof of Theorem 2:** We employ mathematical induction to prove the result as follows:
(1)When n=2
$$\begin{array}{cc} \tilde{\alpha }⊕\tilde{\alpha } & =\left(\left[\frac{2a(1-a)}{1-a^{2}},\frac{2b(1-b)}{1-b^{2}}\right]\right.,\;\;\left.\left[\frac{c^{2}}{c(2-c)},\frac{d^{2}}{d(2-d)}\right]\right)\\ & =\left(\left[\frac{2a}{1+(2-1)a},\frac{2b}{1+(2-1)b}\right],\left[\frac{c}{2-(2-1)c},\frac{d}{2-(2-1)d}\right]\right)\; \end{array}$$Hence, when n=2 the result holds.
(2)Let *n* = m, and assume

$$m⊙\tilde{\alpha }=\underset{m}{\underset{︸}{\tilde{\alpha }⊕\tilde{\alpha }⊕\cdots ⊕\tilde{\alpha }}}=\left(\left[\frac{ma}{1+(m-1)a},\frac{mb}{1+(m-1)b}\right]\right.,\;\;\left.\left[\frac{c}{m-\left(m-1\right)c},\frac{d}{m-\left(m-1\right)d}\right]\right)$$

Then, it follows that
$$\begin{array}{cc} (m+1)⊙\tilde{\alpha } & =m\tilde{\alpha }⊕\tilde{\alpha }=\left(\left[\frac{\left(m+1\right)a}{1+ma},\frac{\left(m+1\right)b}{1+mb}\right],\left[\frac{c}{m+1-mc},\frac{d}{m+1-md}\right]\right)\\ & =\left(\left[\frac{\left(m+1\right)a}{1+\left(m+1-1\right)a},\frac{\left(m+1\right)b}{1+\left(m+1-1\right)b}\right],\right.\left.\left[\frac{c}{m+1-\left(m+1-1\right)c},\frac{d}{m+1-\left(m+1-1\right)}\right]\right) \end{array}$$
Letting n=m+1, we obtain:$$n⊙\tilde{\alpha }=\underset{n}{\underset{︸}{\tilde{\alpha }⊕\tilde{\alpha }⊕\cdots ⊕\tilde{\alpha }}}=\left(\left[\frac{na}{1+(n-1)a},\frac{nb}{1+(n-1)b}\right]\right.,\;\;\left.\left[\frac{c}{n-\left(n-1\right)c},\frac{d}{n-\left(n-1\right)d}\right]\right)$$This completes the proof. □

**Theorem 3.** *For the interval-valued intuitionistic fuzzy number *$\tilde{\alpha }=([a,b],[c,d])$.
${\tilde{\alpha }}^{{}^{⊙n}}=\underset{n}{\underset{︸}{\tilde{\alpha }⊗\tilde{\alpha }⊗\cdots ⊗\tilde{\alpha }}}=\left(\left[\frac{a}{n-(n-1)a},\frac{b}{n-(n-1)b}\right]\right.,\;\;\left.\left[\frac{nc}{1+\left(n-1\right)c},\frac{nd}{1+\left(n-1\right)d}\right]\right)$.

**Proof of Theorem 3:** By Theorems 1 and 2 we obtain
$$\begin{array}{cc} {\tilde{\alpha }}^{{}^{⊙n}} & =\underset{n}{\underset{︸}{\tilde{\alpha }⊗\tilde{\alpha }⊗\cdots ⊗\tilde{\alpha }}}={\left(\underset{n}{\underset{︸}{{\tilde{\alpha }}^{C}⊗{\tilde{\alpha }}^{C}⊗\cdots ⊗{\tilde{\alpha }}^{C}}}\right)}^{C}\\ & =\left(\left[\frac{nc}{1+(n-1)c},\frac{nd}{1+(n-1)d}\right]\right.,\;\;{\left.\left[\frac{a}{n-\left(n-1\right)a},\frac{b}{n-\left(n-1\right)b}\right]\right)}^{C}\\ & =\left(\left[\frac{a}{n-\left(n-1\right)a},\frac{b}{n-\left(n-1\right)b}\right],\left[\frac{nc}{1+\left(n-1\right)c},\frac{nd}{1+\left(n-1\right)d}\right]\right) \end{array}$$
This completes the proof. □

By extending Theorems 2 and 3 to any non-negative real number λ≥0, a new interval-valued intuitionistic fuzzy operation rule can be obtained, as shown in Definition 7.

**Definition 7.** 
*For interval-valued intuitionistic fuzzy numbers*

$\tilde{\alpha }=([a,b],[c,d])$

*and real numbers *

λ≥0

*, we can define the following operations: *

$$\lambda ⊙\tilde{\alpha }=\left(\left[\frac{\lambda a}{1+\left(\lambda -1\right)a},\frac{\lambda b}{1+\left(\lambda -1\right)b}\right],\left[\frac{c}{\lambda -\left(\lambda -1\right)c},\frac{d}{\lambda -\left(\lambda -1\right)d}\right]\right)$$


$${\tilde{\alpha }}^{⊙\lambda }=\left(\left[\frac{a}{\lambda -\left(\lambda -1\right)a},\frac{a}{\lambda -\left(\lambda -1\right)a}\right],\left[\frac{\lambda c}{1+\left(\lambda -1\right)c},\frac{\lambda d}{1+\left(\lambda -1\right)d}\right]\right)$$



**Explanation 3.** When λ>1, the denominators are not equal to 0, so the case where 0≤λ≤1 needs to be focused on. Define the two functions $p(x,y)=\frac{xy}{1+(x-1)y}$, $q(x,y)=\frac{y}{x-(x-1)y}$ where 0≤x≤1, 0≤y≤1. We note that p(x,y) is undefined at (0,1) and q(x,y) is undefined at (0,0).

**Theorem 4.**  
*Let*

$\tilde{\alpha }=([a,b],[c,d])$

*be an interval-valued intuitionistic fuzzy number and*

λ

*be a non-negative real number. Then*

$${\tilde{\alpha }}^{⊙\lambda }={\left(\lambda ⊙{\tilde{\alpha }}^{C}\right)}^{C},\lambda ⊙\tilde{\alpha }={\left({\left({\tilde{\alpha }}^{C}\right)}^{⊙\lambda }\right)}^{C}$$



**Proof of Theorem 4:** Since ${\tilde{\alpha }}^{C}=([c,d],[a,b])$, we obtain
$$\begin{array}{ll} {\left(\lambda ⊙{\tilde{\alpha }}^{C}\right)}^{C} & ={\left(\left[\frac{\lambda c}{1+\left(\lambda -1\right)c},\frac{\lambda d}{1+\left(\lambda -1\right)d}\right],\left[\frac{a}{\lambda -\left(\lambda -1\right)a},\frac{b}{\lambda -\left(\lambda -1\right)b}\right]\right)}^{C}\\ & =\left(\frac{a}{\lambda -\left(\lambda -1\right)a},\frac{b}{\lambda -\left(\lambda -1\right)b},\left[\frac{\lambda c}{1+\left(\lambda -1\right)c},\frac{\lambda d}{1+\left(\lambda -1\right)d}\right]\right)={\tilde{\alpha }}^{⊙\lambda } \end{array}$$Therefore, ${\tilde{\alpha }}^{⊙\lambda }={\left(\lambda ⊙{\tilde{\alpha }}^{C}\right)}^{C}$By the same logic, $\lambda ⊙\tilde{\alpha }={\left({\left({\tilde{\alpha }}^{C}\right)}^{⊙\lambda }\right)}^{C}$This completes the proof. □

**Theorem 5.**  *Let*${\tilde{\alpha }}_{1}=([a_{1},b_{1}],[c_{1},d_{1}])$* and *${\tilde{\alpha }}_{2}=([a_{2},b_{2}],[c_{1},d_{1}]\;$*be interval-valued intuitionistic fuzzy numbers, with *λ, $\lambda_{1}$*, and *$\lambda_{2}$* being non-negative real numbers. Then, the following properties hold:*(1)$\left(\lambda_{1}⊙{\tilde{\alpha }}_{1}\right)⊕\left(\lambda_{2}⊙{\tilde{\alpha }}_{1}\right)=\left(\lambda_{1}+\lambda_{2}\right)⊙{\tilde{\alpha }}_{1}$(2)$\left(\lambda ⊙{\tilde{\alpha }}_{1}\right)⊕\left(\lambda ⊙{\tilde{\alpha }}_{2}\right)=\lambda ⊙\left({\tilde{\alpha }}_{1}⊕{\tilde{\alpha }}_{2}\right)$(3)${\tilde{\alpha }}_{1}{}^{⊙\lambda_{1}}⊗{\tilde{\alpha }}_{1}{}^{⊙\lambda_{2}}={\tilde{\alpha }}_{1}{}^{⊙\left(\lambda_{1}+\lambda_{2}\right)}$(4)${\tilde{\alpha }}_{1}{}^{⊙\lambda }⊗{\tilde{\alpha }}_{2}{}^{⊙\lambda }={\left({\tilde{\alpha }}_{1}⊗{\tilde{\alpha }}_{2}\right)}^{⊙\lambda }$(5)$\lambda_{1}⊙\left(\lambda_{2}⊙{\tilde{\alpha }}_{1}\right)=\left(\lambda_{1}\lambda_{2}\right)⊙{\tilde{\alpha }}_{1}$(6)${\left({\tilde{\alpha }}_{1}{}^{⊙\lambda_{1}}\right)}^{⊙\lambda_{2}}={\tilde{\alpha }}_{1}{}^{⊙\left(\lambda_{1}\lambda_{2}\right)}$

**Proof of Theorem 5:** Theorem 5 follows trivially form Theorems 1–4.Based on Theorem 5 we obtain
$$\left(\lambda_{1}⊙{\tilde{\alpha }}_{1}\right)⊕\cdots ⊕\left(\lambda_{n}⊙{\tilde{\alpha }}_{1}\right)=\overset{n}{\underset{i=1}{⊕}}\left(\lambda_{i}{\tilde{\alpha }}_{1}\right)=\sum_{i=1}^{n}\lambda_{i}⊙{\tilde{\alpha }}_{1}\;\mathrm{and}$$
$$\left({\tilde{\alpha }}_{1}{}^{⊙\lambda_{1}}\right)⊗\cdots ⊗\left({\tilde{\alpha }}_{1}{}^{⊙\lambda_{n}}\right)=\overset{n}{\underset{i=1}{⊗}}\left({\tilde{\alpha }}_{1}{}^{⊙\lambda_{i}}\right)={\tilde{\alpha }}_{1}{}^{⊙\sum_{i=1}^{n}\lambda_{i}}$$In particular, if $\sum_{i=1}^{n}\lambda_{i}=1$, we obtain
$$\left(\lambda_{1}⊙{\tilde{\alpha }}_{1}\right)⊕\left(\lambda_{2}⊙{\tilde{\alpha }}_{1}\right)⊕\cdots ⊕\left(\lambda_{n}⊙{\tilde{\alpha }}_{1}\right)={\tilde{\alpha }}_{1}\;\mathrm{and}$$
$$\left({\tilde{\alpha }}_{1}{}^{⊙\lambda_{1}}\right)⊗\left({\tilde{\alpha }}_{1}{}^{⊙\lambda_{2}}\right)⊗\cdots ⊗\left({\tilde{\alpha }}_{1}{}^{⊙\lambda_{n}}\right)={\tilde{\alpha }}_{1}$$It can be seen from Theorems 1–5 that the new interval-valued intuitionistic fuzzy operation satisfies idempotence, which is essential for the fusion of interval-valued intuitionistic fuzzy information. □

## 4. New Interval-Valued Intuitionistic Fuzzy Weighted Average Operator Model

Based on Theorems 1–5, new interval-valued intuitionistic fuzzy aggregation operations can be defined, and new models of interval-valued intuitionistic fuzzy weighted arithmetic averaging (IVIFWAA) operators and interval-valued intuitionistic fuzzy weighted geometric averaging (IVIFWGA) operators can be proposed.

### 4.1. Interval-Valued Intuitionistic Fuzzy Arithmetic Weighted Average Operator

**Definition 8.**  *Let*${\tilde{\alpha }}_{i}=([a_{i},b_{i}],[c_{i},d_{i}])\;\;(i=1,2,\cdots n)$ *be a set of interval-valued intuitionistic fuzzy numbers, and *$\omega_{i}$* be the weight of *${\tilde{\alpha }}_{i}$*, with *$\omega_{i}\in [0,1]$*, and *$\sum_{i=1}^{n}\omega_{i}=1$*. If the map IVIFWAA: *${\tilde{\Theta }}^{n}\to \tilde{\Theta }$* satisfies *$$\mathrm{IVIFWAA}({\tilde{\alpha }}_{1},{\tilde{\alpha }}_{2},\cdots ,{\tilde{\alpha }}_{n})=\left(\omega_{1}⊙{\tilde{\alpha }}_{1}\right)⊕\left(\omega_{2}⊙{\tilde{\alpha }}_{2}\right)⊕\cdots ⊕\left(\omega_{n}⊙{\tilde{\alpha }}_{n}\right)$$Then, IVIFWAA is an interval-valued intuitionistic fuzzy arithmetic weighted average operator.In particular, when $\omega ={\left(1/n,1/n,\cdots ,1/n\right)}^{T}$, then we obtain
$$\mathrm{IVIFWAA}\left({\tilde{\alpha }}_{1},{\tilde{\alpha }}_{2},\cdots ,{\tilde{\alpha }}_{n}\right)=\left(\frac{1}{n}⊙{\tilde{\alpha }}_{1}\right)⊕\left(\frac{1}{n}⊙{\tilde{\alpha }}_{2}\right)⊕\cdots ⊕\left(\frac{1}{n}⊙{\tilde{\alpha }}_{n}\right)$$By Theorem 5 we obtain
$$\mathrm{IVIFWAA}\left({\tilde{\alpha }}_{1},{\tilde{\alpha }}_{2},\cdots ,{\tilde{\alpha }}_{n}\right)=\frac{1}{n}⊙\left({\tilde{\alpha }}_{1}⊕{\tilde{\alpha }}_{2}⊕\cdots ⊕{\tilde{\alpha }}_{n}\right)$$

**Theorem 6.** 
*Let*

${\tilde{\alpha }}_{i}=([a_{i},b_{i}],[c_{i},d_{i}])\;\;(i=1,2,\cdots n)$

* be a set of interval-valued intuitionistic fuzzy numbers, and *

$\omega_{i}$

* be the weight of *

${\tilde{\alpha }}_{i}$

*, with *

$\omega_{i}\in [0,1]$

*, and *

$\sum_{i=1}^{n}\omega_{i}=1$

*. Then *

$\mathrm{IVIFWAA}\left({\tilde{\alpha }}_{1},{\tilde{\alpha }}_{2},\cdots ,{\tilde{\alpha }}_{n}\right)=\left(\left[a_{A}^{n},b_{A}^{n}\right],\right.\left.\left[c_{A}^{n},d_{A}^{n}\right]\right)$

* is an interval-valued intuitionistic fuzzy number, where *

(1)
$$a_{A}^{n}=1-\frac{\prod_{i=1}^{n}\left(1-a_{i}\right)}{\sum_{j=1}^{n}\left(\omega_{j}\prod_{\begin{array}{l} i=1\\ i\neq j \end{array}}^{n}\left(1-a_{i}\right)\right)-\sum_{j=1}^{n}\left(\left(\omega_{j}-1\right)\prod_{i=1}^{n}\left(1-a_{i}\right)\right)}$$


(2)
$$b_{A}^{n}=1-\frac{\prod_{i=1}^{n}\left(1-b_{i}\right)}{\sum_{j=1}^{n}\left(\omega_{j}\prod_{\begin{array}{l} i=1\\ i\neq j \end{array}}^{n}\left(1-b_{i}\right)\right)-\sum_{j=1}^{n}\left(\left(\omega_{j}-1\right)\prod_{i=1}^{n}\left(1-b_{i}\right)\right)}$$


(3)
$$c_{A}^{n}=\frac{\prod_{i=1}^{n}c_{i}}{\sum_{j=1}^{n}\left(\omega_{j}\prod_{\begin{array}{l} i=1\\ i\neq j \end{array}}^{n}c_{i}\right)-\sum_{j=1}^{n}\left(\left(\omega_{j}-1\right)\prod_{i=1}^{n}c_{i}\right)}$$


(4)
$$d_{A}^{n}=\frac{\prod_{i=1}^{n}d_{i}}{\sum_{j=1}^{n}\left(\omega_{j}\prod_{\begin{array}{l} i=1\\ i\neq j \end{array}}^{n}d_{i}\right)-\sum_{j=1}^{n}\left(\left(\omega_{j}-1\right)\prod_{i=1}^{n}d_{i}\right)}$$



**Proof of Theorem 6:** The result follows from Definition 6 and 7. □

### 4.2. Interval-Valued Intuitionistic Fuzzy Geometric Weighted Average Operator

**Definition 9.** *Let* ${\tilde{\alpha }}_{i}=([a_{i},b_{i}],[c_{i},d_{i}])\;\;(i=1,2,\cdots n)$* be a set of interval-valued intuitionistic fuzzy numbers, and *$\omega_{i}$* be the weight of *${\tilde{\alpha }}_{i}$*, with *$\omega_{i}\in [0,1]$*, and *$\sum_{i=1}^{n}\omega_{i}=1$*. If the map IVIFWGA: *${\tilde{\Theta }}^{n}\to \tilde{\Theta }$* satisfies *$$\mathrm{IVIFWGA}({\tilde{\alpha }}_{1},{\tilde{\alpha }}_{2},\cdots ,{\tilde{\alpha }}_{n})={\tilde{\alpha }}_{1}{}^{⊙\omega_{1}}⊗{\tilde{\alpha }}_{2}{}^{⊙\omega_{2}}⊗\cdots ⊗{\tilde{\alpha }}_{n}{}^{⊙\omega_{n}}$$

Then, IVIFWGA is an interval-valued intuitionistic fuzzy geometric weighted average operator.

In particular, when $\omega ={\left(1/n,1/n,\cdots ,1/n\right)}^{T}$, we obtain
$$\mathrm{IVIFWGA}\left({\tilde{\alpha }}_{1},{\tilde{\alpha }}_{2},\cdots ,{\tilde{\alpha }}_{n}\right)={\tilde{\alpha }}_{1}{}^{⊙\frac{1}{n}}⊗{\tilde{\alpha }}_{2}{}^{⊙\frac{1}{n}}⊗\cdots ⊗{\tilde{\alpha }}_{n}{}^{⊙\frac{1}{n}}$$

By Theorem 5 we obtain
$$\mathrm{IVIFWGA}\left({\tilde{\alpha }}_{1},{\tilde{\alpha }}_{2},\cdots ,{\tilde{\alpha }}_{n}\right)={\left({\tilde{\alpha }}_{1}⊗{\tilde{\alpha }}_{2}⊗\cdots ⊗{\tilde{\alpha }}_{n}\right)}^{⊙\frac{1}{n}}$$

**Theorem 7.** *Let *${\tilde{\alpha }}_{i}=([a_{i},b_{i}],[c_{i},d_{i}])\;\;(i=1,2,\cdots n)$ *be a set of interval-valued intuitionistic fuzzy numbers, and *$\omega_{i}$ *be the weights of *${\tilde{\alpha }}_{i}$*, with *$\omega_{i}\in [0,1]$*, and *$\sum_{i=1}^{n}\omega_{i}=1$*. Then, *$\mathrm{IVIFWGA}\left({\tilde{\alpha }}_{1},{\tilde{\alpha }}_{2},\cdots ,{\tilde{\alpha }}_{n}\right)=\left(\left[a_{G}^{n},b_{G}^{n}\right],\left[c_{G}^{n},d_{G}^{n}\right]\right)$* is an interval-valued intuitionistic fuzzy number, where*(5)$$a_{G}^{n}=\frac{\prod_{i=1}^{n}a_{i}}{\sum_{j=1}^{n}\left(\omega_{j}\prod_{\begin{array}{l} i=1\\ i\neq j \end{array}}^{n}a_{i}\right)-\sum_{j=1}^{n}\left(\left(\omega_{j}-1\right)\prod_{i=1}^{n}a_{i}\right)}$$(6)$$b_{G}^{n}=\frac{\prod_{i=1}^{n}b_{i}}{\sum_{j=1}^{n}\left(\omega_{j}\prod_{\begin{array}{l} i=1\\ i\neq j \end{array}}^{n}b_{i}\right)-\sum_{j=1}^{n}\left(\left(\omega_{j}-1\right)\prod_{i=1}^{n}b_{i}\right)}$$(7)$$c_{G}^{n}=1-\frac{\prod_{i=1}^{n}\left(1-c_{i}\right)}{\sum_{j=1}^{n}\left(\omega_{j}\prod_{\begin{array}{l} i=1\\ i\neq j \end{array}}^{n}\left(1-c_{i}\right)\right)-\sum_{j=1}^{n}\left(\left(\omega_{j}-1\right)\prod_{i=1}^{n}\left(1-c_{i}\right)\right)}$$(8)$$d_{G}^{n}=1-\frac{\prod_{i=1}^{n}\left(1-d_{i}\right)}{\sum_{j=1}^{n}\left(\omega_{j}\prod_{\begin{array}{l} i=1\\ i\neq j \end{array}}^{n}\left(1-d_{i}\right)\right)-\sum_{j=1}^{n}\left(\left(\omega_{j}-1\right)\prod_{i=1}^{n}\left(1-d_{i}\right)\right)}$$

**Proof of Theorem 7:** The result follows directly from Theorem 6.Based on the new interval-valued intuitionistic fuzzy operation properties defined in this document, it can be deduced that IVIFWAA and IVIFWGA operators have the following properties:(1) IdempotenceIf the interval-valued intuitionistic fuzzy numbers are all $\tilde{\alpha }=([a,b],[c,d])$, then
$$\mathrm{IVIFWAA}\left(\tilde{\alpha },\tilde{\alpha },\cdots ,\tilde{\alpha }\right)=\left(\omega_{1}⊙\tilde{\alpha }\right)⊕\left(\omega_{2}⊙\tilde{\alpha }\right)⊕\cdots ⊕\left(\omega_{n}⊙\tilde{\alpha }\right)=\left(\sum_{i=1}^{n}\omega_{i}\right)⊙\tilde{\alpha }=\tilde{\alpha };$$
$$\mathrm{IVIFWGA}\left(\tilde{\alpha },\tilde{\alpha },\cdots ,\tilde{\alpha }\right)=\left({\tilde{\alpha }}^{⊙\omega_{1}}\right)⊕\left({\tilde{\alpha }}^{⊙\omega_{2}}\right)⊕\cdots ⊕\left({\tilde{\alpha }}^{⊙\omega_{n}}\right)={\tilde{\alpha }}^{⊙\sum_{i=1}^{n}\omega_{i}}=\tilde{\alpha }$$(2) Boundedness
$$\mathrm{Let}\;{\tilde{\alpha }}_{\min}=\left(\left[\underset{i=1,2,\cdots ,n}{\min}\{a\},\underset{i=1,2,\cdots ,n}{\min}\{b\}\right],\left[\underset{i=1,2,\cdots ,n}{\max}\{c\},\underset{i=1,2,\cdots ,n}{\max}\{d\}\right]\right),$$
$${\tilde{\alpha }}_{\max}=\left(\left[\underset{i=1,2,\cdots ,n}{\max}\{a\},\underset{i=1,2,\cdots ,n}{\max}\{b\}\right],\left[\underset{i=1,2,\cdots ,n}{\min}\{c\},\underset{i=1,2,\cdots ,n}{\min}\{d\}\right]\right)$$Based on basic operations of monotonicity we obtain:$${\tilde{\alpha }}_{\min}\leq \mathrm{IVIFWAA}\left(\tilde{\alpha },\tilde{\alpha },\cdots ,\tilde{\alpha }\right)\leq {\tilde{\alpha }}_{\max}$$
$${\tilde{\alpha }}_{\min}\leq \mathrm{IVIFWGA}\left(\tilde{\alpha },\tilde{\alpha },\cdots ,{\tilde{\alpha }}_{n}\right)\leq {\tilde{\alpha }}_{\max}$$(3) MonotonicityFor the two sets of interval-valued intuitionistic fuzzy numbers ${\tilde{\alpha }}_{1},{\tilde{\alpha }}_{2},\cdots ,{\tilde{\alpha }}_{n}$ and ${\tilde{\beta }}_{1},{\tilde{\beta }}_{2},\cdots ,{\tilde{\beta }}_{n}$, if ${\tilde{\alpha }}_{i}\leq {\tilde{\beta }}_{i},\forall i\in \{1,2,\cdots ,n\}$, then by the basic operations of monotonicity we obtain
$$\mathrm{IVIFWAA}\left({\tilde{\alpha }}_{1},{\tilde{\alpha }}_{2},\cdots ,{\tilde{\alpha }}_{n}\right)\leq \mathrm{IVIFWAA}\left({\tilde{\beta }}_{1},{\tilde{\beta }}_{2},\cdots ,{\tilde{\beta }}_{n}\right)$$
$$\mathrm{IVIFWGA}\left({\tilde{\alpha }}_{1},{\tilde{\alpha }}_{2},\cdots ,{\tilde{\alpha }}_{n}\right)\leq \mathrm{IVIFWGA}\left({\tilde{\beta }}_{1},{\tilde{\beta }}_{2},\cdots ,{\tilde{\beta }}_{n}\right)$$(4) CommutativitySince the new interval-valued intuitionistic fuzzy operation defined in this paper satisfies commutativity and associativity, IVIFWAA and IVIFWGA also satisfy commutativity. □

## 5. Missile Defense Dynamic Multi-Time Fusion Target Threat Assessment Method Based on IVIFECF-IVIFWA-TOPSIS

### 5.1. IVIFECF-IVIFWA-TOPSIS Assessment Model

The IVIFEC-IVIFWA-TOPSIS assessment model is an effective combination of interval-valued intuitionistic fuzzy set theory and dynamic multiple attribute group decision-making theory. First, the factors that affect the threat assessment of missile defense targets are broken down at multiple levels to establish a threat assessment index system. Then, considering the subjective and objective weights, a comprehensive weight model of target attributes based on IVIFECF is proposed, and the Poisson distribution method is used to solve the weights of the time series to process multi-time situation information. Furthermore, in order to reflect the ambiguity of complex decision-making problems, the decision information is described by interval-valued intuitionistic fuzzy numbers, and a weighted interval-valued intuitionistic fuzzy (WIVIF) decision matrix is constructed. Next, the IVIFWAA/IVIFWGA operator is used to aggregate the decision information of multiple times and multiple decision makers, and the weight of the time series is combined to determine the dynamic multiple time fusion WIVIF decision matrix. Finally, the interval-valued intuitionistic fuzzy numbers are sorted based on the improved TOPSIS method and the threat assessment results are obtained. The flowchart of the IVIFECF-IVIFWA-TOPSIS assessment model is shown in Figure 1.

### 5.2. Description of the Problem

The missile defense dynamic fusion target threat assessment is based on multiple experts quantifying, evaluating, and sorting the attribute values of each incoming target at multiple moments to provide a basis for firepower allocation; that is, a typical dynamic multi-attribute group decision-making problem. Each threat target can be regarded as an alternative plan. Suppose
m incoming targets form a solution set $X=\left\{\left.x_{i}\right|i=1,2,\cdots ,m\right\}$. Target information selects p time points to collect data, and we denote the time series by $T=\left\{t_{k}\left|k=1,2,\cdots ,p\right.\right\}$. Each target has n attributes, and the attribute set is denoted by $C=\left\{c_{j}\left|j=1,2,\cdots ,n\right.\right\}$. The set of decision makers is $D=\left\{D_{s}\left|s=1,2,\cdots ,q\right.\right\}$, and the weight of each decision maker is $\lambda_{s}$, where $\sum_{s=1}^{q}\lambda_{s}=1$. The IVIF decision matrix of the decision maker $D_{s}$ at the moment $t_{k}$ is recorded as $F^{s}(t_{k})={\left(f_{ij}^{s}(t_{k})\right)}_{m\times n}$. That is,
$$F^{s}\left(t_{k}\right)={\left(f_{ij}^{s}\left(t_{k}\right)\right)}_{m\times n}=\begin{array}{cc} & \begin{array}{c} \;c_{1}\;c_{2}\;\cdots \;c_{n} \end{array}\\ \begin{array}{c} x_{1}\\ x_{2}\\ \vdots \\ x_{m} \end{array} & \left(\begin{array}{cccc} f_{11}^{s}\left(t_{k}\right) & f_{12}^{s}\left(t_{k}\right) & \cdots & f_{1n}^{s}\left(t_{k}\right)\\ f_{21}^{s}\left(t_{k}\right) & f_{22}^{s}\left(t_{k}\right) & \cdots & f_{1n}^{s}\left(t_{k}\right)\\ \vdots & \vdots & \ddots & \vdots \\ f_{m1}^{s}\left(t_{k}\right) & f_{m2}^{s}\left(t_{k}\right) & \cdots & f_{mn}^{s}\left(t_{k}\right) \end{array}\right) \end{array}$$


Here, $f_{ij}^{s}(t_{k})=(\left[\mu_{ij}^{L(s)}(t_{k}),\mu_{ij}^{U(s)}(t_{k})\right],\left[v_{ij}^{L(s)}(t_{k}),v_{ij}^{U(s)}(t_{k})\right])$ is the assessment information of the decision maker $D_{s}$ on the target $x_{i}$ at the moment $t_{k}$ based on the attribute $c_{j}$. The condition $\left[\mu_{ij}^{L(s)}(t_{k}),\mu_{ij}^{U(s)}(t_{k})\right]\subset [0,1]$ indicates that the decision maker $D_{s}$ determines that the target $x_{i}$ satisfies the membership interval of the attribute $c_{j}$ at the moment $t_{k}$; $\left[v_{ij}^{L(s)}(t_{k})\right.,\left.v_{ij}^{U(s)}(t_{k})\right]\subset [0,1]$, which indicates that the decision maker $D_{s}$ determines that the target $x_{i}$ satisfies the non-membership interval of the attribute $c_{j}$ at the moment $t_{k}$. Lastly, we note that $\mu_{ij}^{U}(t_{k})+v_{ij}^{U}(t_{k})\leq 1$.

### 5.3. Threat Assessment Index System for Missile Defense Combat Targets

#### 5.3.1. Construction of Threat Assessment Index System

The target characteristics of ballistic missiles can be described by a variety of indicators. Based on the detection information of the incoming ballistic target by the missile defense system sensors, this article starts with target status, target characteristics, and key characteristics, then considers the five aspects of target speed, distance, Radar Cross Section (RCS), interference intensity, and the defense capability of key areas, as secondary index threat factors to construct a missile defense combat target threat assessment index system, as shown in Figure 2.

#### 5.3.2. Quantification of Threat Assessment Indicators

With the attribute characteristics of the target group of ballistic missiles and the types of assessment indicators in mind, this paper uses a semi-S-shaped distribution, a semi-Z-shaped distribution and the G. A. Miller 9-level theory to quantify the threat assessment indicators.

(1) Target distance, RCS attributes

The closer the distance between the target and our defense point, the shorter the time to reach the axis of the route shortcut of our position, and, therefore, the greater the threat of the target. RCS is used as an indicator of stealth performance, the smaller the target RCS, the more likely the radar will find the target, and the greater the threat of the target. Therefore, the target distance and RCS attributes obey the semi-S-shaped distribution. The solution methods of membership and non-membership are shown in Equation (9) and Equation (10), respectively.
(9)$$\mu (x)=\left\{\begin{array}{c} 1,x\leq a\\ \frac{b-x}{b-a},a<x\leq b\\ 0,b<x \end{array}\right.$$
(10)$$v(x)=\left\{\begin{array}{c} 0,x\leq a\\ 2\;\;{\left[\frac{x-a}{b-a}\right]}^{2},a<x\leq \frac{a+b}{2}\\ 1-2\;\;\;{\left[\frac{x-a}{b-a}\right]}^{2},\frac{a+b}{2}<x\leq b\\ 1,b<x \end{array}\right.\;\;$$

(2) Target speed attribute

Flight speed is an important attribute of the target. The faster the flight speed, the smaller the interception time window, and the greater the threat to our strategic location. Thus, the target speed obeys the semi-Z-shaped distribution. Its membership and non-membership degree solution methods are shown in Equation (11) and Equation (12), respectively.
(11)$$\mu (x)=\left\{\begin{array}{c} 0,x\leq a\\ \frac{x-a}{b-a},a<x\leq b\\ 1,b<x \end{array}\right.$$
(12)$$v(x)=\left\{\begin{array}{c} 1,x\leq a\\ 1-2\;\;{\left[\frac{x-a}{b-a}\right]}^{2},a<x\leq \frac{a+b}{2}\\ 2\;\;\;{\left[\frac{x-a}{b-a}\right]}^{2},\frac{a+b}{2}<x\leq b\\ 0,b<x \end{array}\right.$$

(3) Target interference intensity, key defensive ability attributes

The stronger the target interference intensity is, the stronger its penetration capability is, the more difficult it is for the missile defense system to intercept, and the greater the threat of the target. The critical defense capability is closely related to our weapon system combat capability and operational deployment. The stronger the critical defense capability, the smaller the target threat. The above two indicators are qualitative indicators, which can be described quantitatively using G. A. Miller 9-level quantification theory [13]. The corresponding relationship between the quantification results and the interval-valued intuitionistic fuzzy number is shown in Table 1.

### 5.4. Integrated Weight Model of Target Attributes Based on IVIFECF

In the multi-attribute group decision-making process of dynamic multi-time fusion target threat assessment involving multiple decision makers, it is necessary to comprehensively consider the objective weights caused by the differences in the attributes of the targets and the subjective weights caused by the decision maker’s subjective experience and knowledge structure to determine the comprehensive weight of target attributes.

The decision entropy method is a typical objective weight determination method, which can indicate the relative importance of target attributes. The smaller the interval-valued intuitionistic fuzzy entropy, the greater the uncertainty of the information, and the larger the weight of the corresponding solution should be. For existing methods for measuring interval-valued intuitionistic fuzzy entropy when the deviation of the degree of membership and the degree of non-membership are equal, there is inconsistency with the intuitive facts [20,21]. To overcome this problem, this paper proposes an integrated weight model of target attributes based on IVIFECF. We define IVIFECF and verify its effectiveness below.

**Definition 10.** 
*Let*

$\tilde{A}\in IVIFS(X)$

*, then the interval-valued intuitionistic fuzzy entropy *

$E\left(\tilde{A}\right)$

* based on the cosine function can be defined as: *

(13)
$$E\left(\tilde{A}\right)=\frac{1}{m}\sum_{i=1}^{m}\cos\left(\frac{\frac{1}{2}\left|\mu_{\tilde{A}}^{L}(x_{i})-v_{\tilde{A}}^{L}(x_{i})+\mu_{\tilde{A}}^{U}(x_{i})-v_{\tilde{A}}^{U}(x_{i})\right|}{2\left(1+\frac{1}{2}\left(\pi_{\tilde{A}}^{L}(x_{i})+\pi_{\tilde{A}}^{U}(x_{i})\right)\right)}\pi \right)$$



**Theorem 8.** 
*Interval-valued intuitionistic fuzzy entropy*

$E\left(\tilde{A}\right)$

*based on the cosine function has the following properties:*
(1)$E\left(\tilde{A}\right)=0$, if and only if $\tilde{A}$ is a typical set.(2)If $\forall x_{i}\in X,{\tilde{\mu }}_{A}(x_{i})={\tilde{v}}_{A}(x_{i})$, then $E\left(\tilde{A}\right)=1$.(3)$\forall x_{i}\in IVIFS(X)$, $E\left(\tilde{A}\right)=E\left({\tilde{A}}^{C}\right)$.


**Proof of Theorem 8:** 
(1)If $\tilde{A}$ is a typical set, then $\mu_{\tilde{A}}^{L}(x_{i})=\mu_{\tilde{A}}^{U}(x_{i})=1$, $v_{\tilde{A}}^{L}(x_{i})=v_{\tilde{A}}^{U}(x_{i})=0$ and $\pi_{\tilde{A}}^{L}(x_{i})=\pi_{\tilde{A}}^{U}(x_{i})=0$; or $\mu_{\tilde{A}}^{L}(x_{i})=\mu_{\tilde{A}}^{U}(x_{i})=0$, $v_{\tilde{A}}^{L}(x_{i})=v_{\tilde{A}}^{U}(x_{i})=1$ and $\pi_{\tilde{A}}^{L}(x_{i})=\pi_{\tilde{A}}^{U}(x_{i})=0$. Therefore, $E\left(\tilde{A}\right)=0$.(2)When $\mu_{\tilde{A}}^{L}(x_{i})=v_{\tilde{A}}^{L}(x_{i})$ and $\mu_{\tilde{A}}^{U}(x_{i})=v_{\tilde{A}}^{U}(x_{i})$, $E\left(\tilde{A}\right)=\frac{1}{n}\sum_{i=1}^{n}\cos0^{\circ }=1$.(3)${\tilde{A}}^{C}=\left(\left[v_{\tilde{A}}^{L}(x_{i}),v_{\tilde{A}}^{U}(x_{i})\right],\left[\mu_{\tilde{A}}^{L}(x_{i}),\mu_{\tilde{A}}^{U}(x_{i})\right]\right)$, which implies that $E\left(\tilde{A}\right)=E\left({\tilde{A}}^{C}\right)$. □


**Example 1.** Let $\tilde{A}=\left(\left[0.5,0.5\right],\left[0.3,0.3\right]\right)$, $\tilde{B}=\left(\left[0.6,0.6\right]\right.,\left.\left[0.4,0.4\right]\right)$ be two IVIFSs in the universe X. The degree of ambiguity of $\tilde{A}$ is greater than that of $\tilde{B}$, which follows from Equation (13):$$E\left(\tilde{A}\right)=\cos\left(\frac{\left|0.5-0.3+0.5-0.3\right|}{4\left(1+\frac{1}{2}\left(1-0.5-0.3+1-0.5-0.3\right)\right)}\pi \right)=0.966$$
$$E\left(\tilde{B}\right)=\cos\left(\frac{\left|0.6-0.4+0.6-0.4\right|}{4\left(1+\frac{1}{2}\left(1-0.6-0.4+1-0.6-0.4\right)\right)}\pi \right)=0.951$$Thus, $E\left(\tilde{A}\right)>E\left(\tilde{B}\right)$, which is consistent with intuition.

Table 2 shows the calculation results of Example 1 with entropy of IVIFSs in reference [20] and entropy measures of IVIFSs in reference [21]. It can be seen that the interval-valued intuitionistic fuzzy entropy based on the cosine function proposed in this paper can effectively describe the uncertainty of the fuzzy set, and overcome the problem of the existing entropy method being inconsistent with intuition when the deviation of membership degree and non-membership degree is equal.

On this basis, a nonlinear programming model based on minimizing IVIFECF is established to solve the objective weights of target attributes. The steps are as follows:

Step 1: Determine the IVIF decision matrix at the moment $t_{k}$:$$R(t_{k})={\left(\left[\mu_{ij}^{L}(t_{k}),\mu_{ij}^{U}(t_{k})\right],\left[v_{ij}^{L}(t_{k}),v_{ij}^{U}(t_{k})\right]\right)}_{m\times n}$$

Step 2: Using Equation (13), calculate the target attribute interval-valued intuitionistic fuzzy entropy $E_{j}(t_{k})$ at the moment $t_{k}$:$$E_{j}^{}(t_{k})=\frac{1}{n}\sum_{i=1}^{m}\cos\left(\frac{\frac{1}{2}\left|\mu_{ij}^{L}(t_{k})-v_{ij}^{L}(t_{k})+\mu_{ij}^{U}(t_{k})-v_{ij}^{U}(t_{k})\right|}{2\left(1+\frac{1}{2}\left(\pi_{ij}^{L}(t_{k})+\pi_{ij}^{U}(t_{k})\right)\right)}\pi \right)$$

Step 3: Establish a nonlinear programming model based on minimizing IVIFECF, as in Equation (14):(14)$$\begin{array}{l} \min\sum_{j=1}^{n}{\left(\omega_{j}^{\left(1\right)}(t_{k})\right)}^{2}E_{j}(t_{k})\\ s.t.\;\;\;\left\{\begin{array}{l} \sum_{j=1}^{n}\omega_{j}^{\left(1\right)}(t_{k})=1\\ \omega_{j}^{\left(1\right)}(t_{k})\geq 0 \end{array}\right. \end{array}$$
where $\omega_{j}^{\left(1\right)}(t_{k})$ is the weight value of the objective attribute of the target at the moment $t_{k}$.

Step 4: Solve the objective attribute weight of the target $\omega^{\left(1\right)}(t_{k})=(\omega_{1}^{\left(1\right)}(t_{k}),\omega_{2}^{\left(1\right)}(t_{k}),\cdots ,\omega_{n}^{\left(1\right)}(t_{k}))$.

To perform this, establish a Lagrange function for Equation (14),
$$L(\omega ,\lambda )=\sum_{j=1}^{n}{\left(\omega_{j}^{\left(1\right)}(t_{k})\right)}^{2}E_{j}(t_{k})+2\lambda (\sum_{j=1}^{n}\omega_{j}^{\left(1\right)}(t_{k})-1)$$

Take the derivative with respect to $\omega_{j}^{\left(1\right)}(t_{k})$ and λ respectively, and set them equal to 0 to obtain
(15)$$\left\{\begin{array}{l} \frac{\partial L(\omega ,\lambda )}{\partial \omega_{j}^{\left(1\right)}(t_{k})}=2\omega_{j}^{\left(1\right)}(t_{k})\cdot E_{j}^{}(t_{k})+2\lambda =0\\ \frac{\partial L(\omega ,\lambda )}{\partial \lambda }=2(\sum_{j=1}^{n}\omega_{j}^{\left(1\right)}(t_{k})-1)=0 \end{array}\right.$$

Solve Equation (15) to obtain the objective attribute weight of the target at the moment $t_{k}$ to obtain
(16)$$\omega_{j}^{\left(1\right)}(t_{k})=\frac{{\left(E_{j}(t_{k})\right)}^{-1}}{\sum_{j=1}^{n}{\left(E_{j}(t_{k})\right)}^{-1}}$$

Step 5: Suppose the subjective attribute weight vector provided by the decision-maker $D_{s}$ is $\Omega =\{\omega_{j}^{s(2)}\left|j=1,2,\cdots ,n\right.,s=1,2,\cdots ,q\}$, where $\omega_{j}^{s(2)}\in [0,1]$ and $\sum_{j=1}^{n}\omega_{j}^{s(2)}=1$. We use the product rule to obtain the comprehensive weight of the target attribute of the decision-maker $D_{s}$ at the moment $t_{k}$, as shown in Equation (17)
(17)$$\omega_{j}^{s}(t_{k})=\frac{\omega_{j}^{\left(1\right)}(t_{k})\omega_{j}^{s(2)}(t_{k})}{\sum_{j=1}^{m}\omega_{j}^{\left(1\right)}(t_{k})\omega_{j}^{s(2)}(t_{k})}$$

### 5.5. Time Series Weight Model Based on Poisson Distribution Method

In missile defense operations, the threat level of the target will dynamically change with time and the battlefield situation. In order to improve the accuracy of target threat assessment, it is not only necessary to consider the target information at the current moment, but also to take into account different points in a time series. In actual missile defense combat, the closer to the current moment, the greater the impact of target information acquisition on the results of the threat assessment. Therefore, this paper selects the target information collected at the current time p and the previous time p−1, and uses the Poisson distribution method to solve the time series weight $\eta =(\eta_{1},\eta_{2},\cdots ,\eta_{p})$ in the inverse form, as shown in Equation (18):(18)$$\eta_{k}=\frac{k!/\varphi^{k}}{\sum_{j=1}^{m}\left(k!/\varphi^{k}\right)}$$
where $\eta_{k}\geq 0$ satisfies $\sum_{k=1}^{p}\eta_{k}=0$ and 0<φ<2.

### 5.6. Multi-Source Information Aggregation Based on IVIFWAA/IVIFWGA Operators

The missile defense dynamic target threat assessment in an uncertain environment needs to be based on a comprehensive weight calculation result of the target attribute. Starting with the WIVIF decision matrix $R_{s}(t_{k})$ of the decision maker $D_{s}$ at the moment $t_{k}$, we perform interval-valued intuitionistic fuzzy information aggregations on the decision information of multiple target attributes, multiple moments, and multiple experts. The IVIFWAA and IVIFWGA operators proposed in this paper have algebraic characteristics, such as idempotence, boundedness, monotonicity, and commutativity, and they have important advantages in information fusion. Therefore, this article uses IVIFWAA/IVIFWGA operators to aggregate multi-source information.

Suppose the WIVIF decision matrix of the decision maker $D_{s}$ at time $t_{k}$ is recorded as $R^{s}(t_{k})={\left(r_{ij}^{s}(t_{k})\right)}_{m\times n}$. That is,
$$R^{s}\left(t_{k}\right)={\left(r_{ij}^{s}\left(t_{k}\right)\right)}_{m\times n}=\begin{array}{cc} & \begin{array}{c} \;c_{1}\;c_{2}\;\cdots \;c_{n} \end{array}\\ \begin{array}{c} x_{1}\\ x_{2}\\ \vdots \\ x_{m} \end{array} & \left(\begin{array}{cccc} r_{11}^{s}\left(t_{k}\right) & r_{12}^{s}\left(t_{k}\right) & \cdots & r_{1n}^{s}\left(t_{k}\right)\\ r_{21}^{s}\left(t_{k}\right) & r_{22}^{s}\left(t_{k}\right) & \cdots & r_{1n}^{s}\left(t_{k}\right)\\ \vdots & \vdots & \ddots & \vdots \\ r_{m1}^{s}\left(t_{k}\right) & r_{m2}^{s}\left(t_{k}\right) & \cdots & r_{mn}^{s}\left(t_{k}\right) \end{array}\right) \end{array}$$
where
(19)$$r_{ij}^{s}(t_{k})=([1-{\left(1-\mu_{ij}^{L(s)}(t_{k})\right)}^{\omega_{j}^{s}(t_{k})},1-{\left(1-\mu_{ij}^{U(s)}(t_{k})\right)}^{\omega_{j}^{s}(t_{k})}],[v_{ij}^{L(s)}{\left(t_{k}\right)}^{\omega_{j}^{s}(t_{k})},v_{ij}^{U(s)}{\left(t_{k}\right)}^{\omega_{j}^{s}(t_{k})}])$$

After determining the WIVIF decision matrix, use the IVIFWAA/IVIFWGA operator for each decision maker to aggregate the assessment results of each solution on all attributes, and then the assessment results of each decision maker for all solutions can be obtained:
$$R\left(t_{k}\right)={\left(r_{i}^{s}\left(t_{k}\right)\right)}_{m\times q}=\begin{array}{cc} & \begin{array}{c} D_{1}\;D_{2}\;\cdots \;D_{Q} \end{array}\\ \begin{array}{c} x_{1}\\ x_{2}\\ \vdots \\ x_{m} \end{array} & \left(\begin{array}{cccc} r_{1}^{1}\left(t_{k}\right) & r_{1}^{2}\left(t_{k}\right) & \cdots & r_{1}^{q}\left(t_{k}\right)\\ r_{2}^{1}\left(t_{k}\right) & r_{2}^{2}\left(t_{k}\right) & \cdots & r_{2}^{q}\left(t_{k}\right)\\ \vdots & \vdots & \ddots & \vdots \\ r_{m}^{1}\left(t_{k}\right) & r_{m}^{2}\left(t_{k}\right) & \cdots & r_{m}^{q}\left(t_{k}\right) \end{array}\right) \end{array}$$


If we use the IVIFWAA operator for aggregation, we obtain
$$r_{i}^{s}(t_{k})=(\left[\mu_{i}^{L(s)}(t_{k}),\mu_{i}^{U(s)}(t_{k})\right],\left[v_{i}^{L(s)}(t_{k}),v_{i}^{U(s)}(t_{k})\right])=\mathrm{IVIFWAA}\left(r_{i1}^{s}(t_{k}),r_{i2}^{s}(t_{k}),\cdots ,r_{in}^{s}(t_{k})\right)$$
where
(20)$$\mu_{i}^{L(s)}(t_{k})=1-\frac{\prod_{j=1}^{n}\left(1-\mu_{ij}^{L(s)}(t_{k})\right)}{\sum_{j=1}^{n}\left(\omega_{j}^{s}(t_{k})\prod_{\begin{array}{l} i=1\\ i\neq j \end{array}}^{n}\left(1-\mu_{ij}^{L(s)}(t_{k})\right)\right)-\sum_{j=1}^{n}\left(\left(\omega_{j}^{s}(t_{k})-1\right)\prod_{j=1}^{n}\left(1-\mu_{ij}^{L(s)}(t_{k})\right)\right)}$$
(21)$$\mu_{i}^{U(s)}(t_{k})=1-\frac{\prod_{j=1}^{n}\left(1-\mu_{ij}^{U(s)}(t_{k})\right)}{\sum_{j=1}^{n}\left(\omega_{j}^{s}(t_{k})\prod_{\begin{array}{l} i=1\\ i\neq j \end{array}}^{n}\left(1-\mu_{ij}^{U(s)}(t_{k})\right)\right)-\sum_{j=1}^{n}\left(\left(\omega_{j}^{s}(t_{k})-1\right)\prod_{j=1}^{n}\left(1-\mu_{ij}^{U(s)}(t_{k})\right)\right)}$$
(22)$$v_{i}^{L(s)}(t_{k})=\frac{\prod_{j=1}^{n}v_{ij}^{L(s)}(t_{k})}{\sum_{j=1}^{n}\left(\omega_{j}^{s}(t_{k})\prod_{\begin{array}{l} i=1\\ i\neq j \end{array}}^{n}v_{ij}^{L(s)}(t_{k})\right)-\sum_{j=1}^{n}\left(\left(\omega_{j}^{s}(t_{k})-1\right)\prod_{j=1}^{n}v_{ij}^{L(s)}(t_{k})\right)}$$
(23)$$v_{i}^{U(s)}(t_{k})=\frac{\prod_{j=1}^{n}v_{ij}^{U(s)}(t_{k})}{\sum_{j=1}^{n}\left(\omega_{j}^{s}(t_{k})\prod_{\begin{array}{l} i=1\\ i\neq j \end{array}}^{n}v_{ij}^{U(s)}(t_{k})\right)-\sum_{j=1}^{n}\left(\left(\omega_{j}^{s}(t_{k})-1\right)\prod_{j=1}^{n}v_{ij}^{U(s)}(t_{k})\right)}$$

If we use the IVIFWGA operator for aggregation, we obtain:$$r_{i}^{s}(t_{k})=(\left[\mu_{i}^{L(s)}(t_{k}),\mu_{i}^{U(s)}(t_{k})\right],\left[v_{i}^{L(s)}(t_{k}),v_{i}^{U(s)}(t_{k})\right])=\mathrm{IVIFWGA}\left(r_{i1}^{s}(t_{k}),r_{i2}^{s}(t_{k}),\cdots ,r_{in}^{s}(t_{k})\right)$$
where
(24)$$\mu_{i}^{L(s)}(t_{k})=\frac{\prod_{j=1}^{n}\mu_{ij}^{L(s)}(t_{k})}{\sum_{j=1}^{n}\left(\omega_{j}^{s}(t_{k})\prod_{\begin{array}{l} i=1\\ i\neq j \end{array}}^{n}\mu_{ij}^{L(s)}(t_{k})\right)-\sum_{j=1}^{n}\left(\left(\omega_{j}^{s}(t_{k})-1\right)\prod_{j=1}^{n}\mu_{ij}^{L(s)}(t_{k})\right)}$$
(25)$$\mu_{i}^{U(s)}(t_{k})=\frac{\prod_{j=1}^{n}\mu_{ij}^{U(s)}(t_{k})}{\sum_{j=1}^{n}\left(\omega_{j}^{s}(t_{k})\prod_{\begin{array}{l} i=1\\ i\neq j \end{array}}^{n}\mu_{ij}^{U(s)}(t_{k})\right)-\sum_{j=1}^{n}\left(\left(\omega_{j}^{s}(t_{k})-1\right)\prod_{j=1}^{n}\mu_{ij}^{U(s)}(t_{k})\right)}$$
(26)$$v_{i}^{L(s)}(t_{k})=1-\frac{\prod_{j=1}^{n}\left(1-v_{ij}^{L(s)}(t_{k})\right)}{\sum_{j=1}^{n}\left(\omega_{j}^{s}(t_{k})\prod_{\begin{array}{l} i=1\\ i\neq j \end{array}}^{n}\left(1-v_{ij}^{L(s)}(t_{k})\right)\right)-\sum_{j=1}^{n}\left(\left(\omega_{j}^{s}(t_{k})-1\right)\prod_{j=1}^{n}\left(1-v_{ij}^{L(s)}(t_{k})\right)\right)}$$
(27)$$v_{i}^{U(s)}(t_{k})=1-\frac{\prod_{j=1}^{n}\left(1-v_{ij}^{U(s)}(t_{k})\right)}{\sum_{j=1}^{n}\left(\omega_{j}^{s}(t_{k})\prod_{\begin{array}{l} i=1\\ i\neq j \end{array}}^{n}\left(1-v_{ij}^{U(s)}(t_{k})\right)\right)-\sum_{j=1}^{n}\left(\left(\omega_{j}^{s}(t_{k})-1\right)\prod_{j=1}^{n}\left(1-v_{ij}^{U(s)}(t_{k})\right)\right)}$$

Furthermore, the IVIFWAA/IVIFWGA operator is used to aggregate the assessment results of all decision makers in the same solution, and $Z_{i}(i=1,2,\cdots ,m)$ is used to represent the final assessment result of the solution $X_{i}$.

If we use the IVIFWAA operator for aggregation, we obtain
$$Z_{i}(t_{k})=(\left[\mu_{i}^{L}(t_{k}),\mu_{i}^{U}(t_{k})\right],\left[v_{i}^{L}(t_{k}),v_{i}^{U}(t_{k})\right])=\mathrm{IVIFWAA}\left(r_{i}^{1}(t_{k}),r_{i}^{2}(t_{k}),\cdots ,r_{i}^{q}(t_{k})\right)$$
where
(28)$$\mu_{i}^{L}(t_{k})=1-\frac{\prod_{s=1}^{q}\left(1-\mu_{i}^{L(s)}(t_{k})\right)}{\sum_{j=1}^{q}\left(\lambda_{j}\prod_{\begin{array}{l} s=1\\ j\neq s \end{array}}^{q}\left(1-\mu_{i}^{L(s)}(t_{k})\right)\right)-\sum_{s=1}^{q}\left(\left(\lambda_{s}-1\right)\prod_{s=1}^{q}\left(1-\mu_{i}^{L(s)}(t_{k})\right)\right)}$$
(29)$$\mu_{i}^{U}(t_{k})=1-\frac{\prod_{s=1}^{q}\left(1-\mu_{i}^{U(s)}(t_{k})\right)}{\sum_{j=1}^{q}\left(\lambda_{j}\prod_{\begin{array}{l} s=1\\ j\neq s \end{array}}^{q}\left(1-\mu_{i}^{U(s)}(t_{k})\right)\right)-\sum_{s=1}^{q}\left(\left(\lambda_{s}-1\right)\prod_{s=1}^{q}\left(1-\mu_{i}^{U(s)}(t_{k})\right)\right)}$$
(30)$$v_{i}^{L}(t_{k})=\frac{\prod_{s=1}^{q}v_{i}^{L(s)}(t_{k})}{\sum_{j=1}^{q}\left(\lambda_{j}\prod_{\begin{array}{l} s=1\\ s\neq j \end{array}}^{q}v_{i}^{L(s)}(t_{k})\right)-\sum_{s=1}^{q}\left(\left(\lambda_{s}-1\right)\prod_{s=1}^{q}v_{i}^{L(s)}(t_{k})\right)}$$
(31)$$v_{i}^{U}(t_{k})=\frac{\prod_{s=1}^{q}v_{i}^{U(s)}(t_{k})}{\sum_{j=1}^{q}\left(\lambda_{j}\prod_{\begin{array}{l} s=1\\ s\neq j \end{array}}^{q}v_{i}^{U(s)}(t_{k})\right)-\sum_{s=1}^{q}\left(\left(\lambda_{s}-1\right)\prod_{s=1}^{q}v_{i}^{U(s)}(t_{k})\right)}$$

If we use the IVIFWGA operator for aggregation, we obtain
$$Z_{i}(t_{k})=(\left[\mu_{i}^{L}(t_{k}),\mu_{i}^{U}(t_{k})\right],\left[v_{i}^{L}(t_{k}),v_{i}^{U}(t_{k})\right])=\mathrm{IVIFWGA}\left(r_{i}^{1}(t_{k}),r_{i}^{2}(t_{k}),\cdots ,r_{i}^{q}(t_{k})\right)$$
where
(32)$$\mu_{i}^{L}(t_{k})=\frac{\prod_{s=1}^{q}\mu_{i}^{L(s)}(t_{k})}{\sum_{j=1}^{q}\left(\lambda_{j}\prod_{\begin{array}{l} s=1\\ s\neq j \end{array}}^{q}\mu_{i}^{L(s)}(t_{k})\right)-\sum_{s=1}^{q}\left(\left(\lambda_{s}-1\right)\prod_{s=1}^{q}\mu_{i}^{L(s)}(t_{k})\right)}$$
(33)$$\mu_{i}^{U}(t_{k})=\frac{\prod_{s=1}^{q}\mu_{i}^{U(s)}(t_{k})}{\sum_{j=1}^{q}\left(\lambda_{j}\prod_{\begin{array}{l} s=1\\ s\neq j \end{array}}^{q}\mu_{i}^{U(s)}(t_{k})\right)-\sum_{s=1}^{q}\left(\left(\lambda_{s}-1\right)\prod_{s=1}^{q}\mu_{i}^{U(s)}(t_{k})\right)}$$
(34)$$v_{i}^{L}(t_{k})=1-\frac{\prod_{s=1}^{q}\left(1-v_{i}^{L(s)}(t_{k})\right)}{\sum_{j=1}^{q}\left(\lambda_{j}\prod_{\begin{array}{l} s=1\\ j\neq s \end{array}}^{q}\left(1-v_{i}^{L(s)}(t_{k})\right)\right)-\sum_{s=1}^{q}\left(\left(\lambda_{s}-1\right)\prod_{s=1}^{q}\left(1-v_{i}^{L(s)}(t_{k})\right)\right)}$$
(35)$$v_{i}^{U}(t_{k})=1-\frac{\prod_{s=1}^{q}\left(1-v_{i}^{U(s)}(t_{k})\right)}{\sum_{j=1}^{q}\left(\lambda_{j}\prod_{\begin{array}{l} s=1\\ j\neq s \end{array}}^{q}\left(1-v_{i}^{U(s)}(t_{k})\right)\right)-\sum_{s=1}^{q}\left(\left(\lambda_{s}-1\right)\prod_{s=1}^{q}\left(1-v_{i}^{U(s)}(t_{k})\right)\right)}$$

Using Equations (20)–(35), multi-target attributes, multi-decision makers, and multi-time interval-valued intuitionistic fuzzy information aggregation can be realized, laying the foundation for dynamic fusion target threat assessment.

### 5.7. Ordering Method of Interval-Valued Intuitionistic Fuzzy Numbers Based on Improved TOPSIS

The data obtained from aggregation using the IVIFWAA/IVIFWGA operator are still interval-valued intuitionistic fuzzy numbers. Thus, to obtain the threat ordering of the incoming target, it is necessary to compare the magnitudes of the interval-valued intuitionistic fuzzy numbers. Tan et al. [28] verifies that the interval-valued intuitionistic fuzzy number ordering method based on TOPSIS is highly useful for classification. In order to improve the ability to distinguish and differentiate between decision-making results, this paper, based on results in [28], considers the influence of hesitation on distance measurement and proposes an improved interval-valued intuitionistic fuzzy number distance measurement model.

**Definition 11.** 
*Let*

$\tilde{\alpha }=([a_{1},b_{1}],[c_{1},d_{1}])\;\;$

*,*

$\tilde{\beta }=([a_{2},b_{2}],\;\;[c_{2},d_{2}])$

* be two interval-valued intuitionistic fuzzy numbers. The improved interval-valued intuitionistic fuzzy number distance measurement model can be defined as*

$$d(\tilde{\alpha },\tilde{\beta })=\;\sqrt{\frac{1}{4}[{\left(a_{1}-a_{2}\right)}^{2}+{\left(b_{1}-b_{2}\right)}^{2}+{\left(c_{1}-c_{2}\right)}^{2}+{\left(d_{1}-d_{2}\right)}^{2}+{\left(a_{1}+c_{1}-a_{2}-c_{2}\right)}^{2}\;+{\left(b_{1}+d_{1}-b_{2}-d_{2}\right)}^{2}]}$$



In the dynamic multi-time fusion WIVIF decision matrix $H={\left(\left[\mu_{ik}^{L},\mu_{ik}^{U}\right],\left[v_{ik}^{L},v_{ik}^{U}\right]\right)}_{m\times p}$, the positive ideal solution is the solution with the greatest threat degree among all targets, and the negative ideal solution is the solution with the least threat degree.

The positive ideal solution of H is
$$h^{+}=\left(([\mu_{1}^{L+},\mu_{1}^{U+}],[v_{1}^{L+},v_{1}^{U+}]\;),([\mu_{2}^{L+},\mu_{2}^{U+}],\;[v_{2}^{L+},v_{2}^{U+}]),\cdots ,\right.\;\;\left.\left([\mu_{p}^{L+},\mu_{p}^{U+}],[v_{p}^{L+},v_{p}^{U+}]\;\right)\right)$$
where
$$\mu_{k}^{L+}=\underset{1\leq i\leq m}{\max}\left\{\mu_{ik}^{L}\right\},\mu_{k}^{U+}=\underset{1\leq i\leq m}{\max}\left\{\mu_{ik}^{U}\right\},v_{k}^{L+}=\underset{1\leq i\leq m}{\min}\left\{v_{ik}^{L}\right\},v_{k}^{U+}=\underset{1\leq i\leq m}{\min}\left\{v_{ik}^{U}\right\}$$

The negative ideal solution of H is
$$h^{-}=\left(([\mu_{1}^{L-},\mu_{1}^{U-}],[v_{1}^{L-},v_{1}^{U-}]\;),([\mu_{2}^{L-},\mu_{2}^{U-}],\;[v_{2}^{L-},v_{2}^{U-}]),\cdots ,\right.\;\;\;\;\left.\left([\mu_{p}^{L-},\mu_{p}^{U-}],[v_{p}^{L-},v_{p}^{U-}]\;\right)\right)$$
where
$$\mu_{k}^{L-}=\underset{1\leq i\leq m}{\min}\left\{\mu_{ik}^{L}\right\},\mu_{k}^{U-}=\underset{1\leq i\leq m}{\min}\left\{\mu_{ik}^{U}\right\},v_{k}^{L-}=\underset{1\leq i\leq m}{\max}\left\{v_{ik}^{L}\right\},v_{k}^{U-}=\underset{1\leq i\leq m}{\max}\left\{v_{ik}^{U}\right\}$$

According to Definition (11), the respective distances between each target xi and the positive and negative ideal solutions of the dynamic multi-time fusion WIVIF decision matrix are $d_{i}^{+}=d({\tilde{h}}_{i},h^{+})$ and $d_{i}^{-}=d({\tilde{h}}_{i},h^{-})$, as shown below:(36)$$d_{i}^{+}=\sqrt{\sum_{k=1}^{p}\left[\frac{1}{4}[{\left(\mu_{ik}^{L}-\mu_{k}^{L+}\right)}^{2}+{\left(\mu_{ik}^{U}-\mu_{k}^{U+}\right)}^{2}+{\left(v_{ik}^{L}-v_{k}^{L+}\right)}^{2}+{\left(v_{ik}^{U}-v_{k}^{U+}\right)}^{2}+{\left(\mu_{ik}^{L}+v_{ik}^{L}-\mu_{k}^{L+}-v_{k}^{L+}\right)}^{2}\;+{\left(\mu_{ik}^{U}+v_{ik}^{U}-\mu_{k}^{U+}-v_{k}^{U+}\right)}^{2}]\right]}$$
(37)$$d_{i}^{-}=\sqrt{\sum_{k=1}^{p}\left[\frac{1}{4}[{\left(\mu_{ik}^{L}-\mu_{k}^{L-}\right)}^{2}+{\left(\mu_{ik}^{U}-\mu_{k}^{U-}\right)}^{2}+{\left(v_{ik}^{L}-v_{k}^{L-}\right)}^{2}+{\left(v_{ik}^{U}-v_{k}^{U-}\right)}^{2}+{\left(\mu_{ik}^{L}+v_{ik}^{L}-\mu_{k}^{L-}-v_{k}^{L-}\right)}^{2}\;+{\left(\mu_{ik}^{U}+v_{ik}^{U}-\mu_{k}^{U-}-v_{k}^{U-}\right)}^{2}]\right]}$$

Based on the TOPSIS principle, the relative closeness of the target xi is the threat of the target, as shown below:(38)$$\zeta_{i}=\frac{d_{i}^{-}}{d_{i}^{-}+d_{i}^{+}}$$

### 5.8. Algorithm Flow

The specific steps of the missile defense dynamic fusion target threat assessment method based on IVIFECF-IVIFWA-TOPSIS are as follows:

Step 1: Construct a threat assessment index system and use Equations (9)–(12) to quantify each threat assessment index.

Step 2: Calculate the comprehensive weight $\omega^{s}(t_{k})=(\omega_{1}^{s}(t_{k}),\omega_{2}^{s}(t_{k}),\cdots ,\omega_{n}^{s}(t_{k}))$ of the target attribute of the decision maker $D_{s}$ at the moment $t_{k}$ according to Equations (13)–(17).

Step 3: Calculate the WIVIF decision matrix $R^{s}(t_{k})={\left(r_{ij}^{s}(t_{k})\right)}_{m\times n}$ of the decision maker $D_{s}$ at the moment $t_{k}$ according to Equation (19).

Step 4: Use the IVIFWAA/IVIFWGA operator to aggregate the assessment results of each solution for each attribute and calculate the WIVIF decision matrix $R(t_{k})={\left(r_{i}^{s}(t_{k})\right)}_{m\times q}$ of all decision makers at the moment $t_{k}$ according to Equations (20)–(27).

Step 5: Use the IVIFWAA/IVIFWGA operator to aggregate the decision information of q decision makers and obtain the intuitionistic fuzzy value $Z_{i}(t_{k})$ of the target interval at a single moment according to Equations (28)–(35).

Step 6: Determine the weight $\eta =(\eta_{1},\eta_{2},\cdots ,\eta_{p})$ of the time series according to Equation (18) and construct a dynamic multi-time fusion WIVIF matrix $H={\left[h_{ik}\right]}_{m\times p}$ from time $t_{1}$ to $t_{p}$, where $h_{ik}=\eta_{k}⊙Z_{i}(t_{k})$.

Step 7: Obtain $H={\left[h_{ik}\right]}_{m\times p}$ positive and negative ideal solutions, and calculate the distances $d_{i}^{+}$ and $d_{i}^{-}$ between the target $x_{i}$ and the H positive and negative ideal solutions, as well as the degree of threat $\zeta_{i}$ according to Equations (36)–(38), to obtain the final threat assessment result of the target.

## 6. Simulation and Result Analysis

Suppose that in a certain missile defense exercise, the sensors of the missile defense system observed four batches of incoming targets $x_{i}(i=1,2,3,4)$. After obtaining the attribute data in the form of the interval values of the target at three consecutive times $t_{k}(k=1,2,3)$, three experts $D_{s}(s=1,2,3)$ are assigned to determine the five attributes $c_{j}=(j=1,2,3,4,5)$ of speed, distance, RCS, interference strength, and defense capability so that the target threat degree can be evaluated.

### 6.1. Algorithm Feasibility Test and Analysis

Step 1: Use Equations (9)–(12) to quantify each threat assessment index, and apply the method proposed in [29] to convert the interval value into an interval-valued intuitionistic fuzzy value, as shown in Table 3.

Step 2: Solve the comprehensive weight of the target attribute based on IVIFECF. First, the interval intuitionistic fuzzy entropy of each target speed attributes at $t_{1}$ is calculated according to Equation (13).
$$E_{1}(t_{1})=\cos\left(\frac{\frac{1}{2}\left|0.52-0.37+0.55-0.41\right|}{2\left(1+\frac{1}{2}\left(1-0.52-0.37+1-0.55-0.41\right)\right)}\pi \right)=0.9776$$
$$E_{2}(t_{1})=\cos\left(\frac{\frac{1}{2}\left|0.44-0.48+0.49-0.5\right|}{2\left(1+\frac{1}{2}\left(1-0.44-0.48+1-0.49-0.5\right)\right)}\pi \right)=0.9992$$
$$E_{3}(t_{1})=\cos\left(\frac{\frac{1}{2}\left|0.47-0.36+0.53-0.42\right|}{2\left(1+\frac{1}{2}\left(1-0.47-0.36+1-0.53-0.42\right)\right)}\pi \right)=0.9879$$
$$E_{4}(t_{1})=\cos\left(\frac{\frac{1}{2}\left|0.56-0.27+0.62-0.38\right|}{2\left(1+\frac{1}{2}\left(1-0.56-0.27+1-0.62-0.38\right)\right)}\pi \right)=0.9273$$
$$E(t_{1})=\frac{1}{4}\sum_{i=1}^{4}(0.9776+0.9992+0.9879+0.9273)=0.9730$$

Then, the interval intuitionistic fuzzy entropy of each attribute of the target at $t_{1}-t_{3}$ is shown in Table 4.

By Equation (16), the objective weights of the target attributes at $t_{1}-t_{3}$ are:$$\omega^{\left(1\right)}(t_{1})=(0.1784,0.1991,0.2466,0.1907,0.1852)$$
$$\omega^{\left(1\right)}(t_{2})=(0.1642,0.1901,0.2646,0.2065,0.1746)$$
$$\omega^{\left(1\right)}(t_{3})=(0.1483,0.1987,0.2503,0.2245,0.1782{)}_{\;}$$

Suppose the weights of the three experts participating in decision-making $D_{1}$, $D_{2}$, $D_{3}$ are respectively λ=(0.33,0.34,0.33), and the subjective weights of the target attributes given by the decision-makers $D_{1}$, $D_{2}$, $D_{3}$ are, respectively:$$\omega_{1}^{\left(2\right)}=[0.26,0.28,0.16,0.20,0.10]$$
$$\omega_{2}{}^{\left(2\right)}=[0.23,0.30,0.12,0.17,0.08]$$
$$\omega_{3}{}^{\left(2\right)}=[0.31,0.20,0.17,0.18,0.14]$$

After obtaining the objective and subjective weights of the target attributes from $t_{1}-t_{3}$, the comprehensive weights of the target attributes of experts $D_{1}$, $D_{2}$, $D_{3}$ at $t_{1}-t_{3}$ can be obtained by Equation (17):$$\omega_{1}(t_{1})=(0.2340,0.2812,0.1990,0.1924,0.0934)$$
$$\omega_{2}(t_{1})=(0.2311,0.3363,0.1666,0.1826,0.0834)$$
$$\omega_{3}(t_{1})=(0.2803,0.2018,0.2125,0.1740,0.1314)$$
$$\omega_{1}(t_{2})=(0.2167,0.2702,0.2149,0.2096,0.0886)$$
$$\omega_{2}(t_{2})=(0.2151,0.3247,0.1808,0.1999,0.0795)$$
$$\omega_{3}(t_{2})=(0.2603,0.1945,0.2301,0.1901,0.1250)$$
$$\omega_{1}(t_{3})=(0.1958,0.2825,0.2033,0.2280,0.0904)$$
$$\omega_{2}(t_{3})=(0.1936,0.3384,0.1705,0.2166,0.0809)$$
$$\omega_{3}(t_{3})=(0.2375,0.2052,0.2198,0.2087,0.1288)$$

Step 3: Using Equation (19), we obtain the WIVIF decision matrix
$R^{s}(t_{k})$ of experts $D_{1}$, $D_{2}$, $D_{3}$ at $t_{1}-t_{3}$.
$$R^{1}\left(t_{1}\right)=\begin{array}{cc} & \begin{array}{c} \;c_{1}\;c_{2}\;c_{3}\;c_{4}\;c_{5} \end{array}\\ \begin{array}{c} x_{1}\\ x_{2}\\ x_{3}\\ x_{4} \end{array} & \left(\begin{array}{ccccc} \left([0.16,0.17],[0.79,0.81]\right) & \left([0.26,0.28],[0.67,0.70]\right) & \left([0.21,0.22],[0.75,0.77]\right) & \left([0.21,0.27],[0.73,0.73]\right) & \left([0.05,0.06],[0.89,0.92]\right)\\ \left([0.13,0.15],[0.84,0.85]\right) & \left([0.23,0.26],[0.71,0.73]\right) & \left([0.22,0.24],[0.73,0.76]\right) & \left([0.12,0.12],[0.79,0.84]\right) & \left([0.08,0.11],[0.86,0.89]\right)\\ \left([0.14,0.16],[0.79,0.82]\right) & \left([0.22,0.24],[0.74,0.76]\right) & \left([0.24,0.26],[0.71,0.73]\right) & \left([0.16,0.21],[0.73,0.79]\right) & \left([0.02,0.03],[0.94,0.95]\right)\\ \left([0.17,0.20],[0.74,0.80]\right) & \left([0.25,0.27],[0.69,0.72]\right) & \left([0.23,0.27],[0.69,0.72]\right) & \left([0.09,0.12],[0.79,0.84]\right) & \left([0.03,0.05],[0.92,0.94]\right) \end{array}\right) \end{array}$$
$$R^{2}\left(t_{1}\right)=\begin{array}{cc} & \begin{array}{c} \;c_{1}\;c_{2}\;c_{3}\;c_{4}\;c_{5} \end{array}\\ \begin{array}{c} x_{1}\\ x_{2}\\ x_{3}\\ x_{4} \end{array} & \left(\begin{array}{ccccc} \left([0.16,0.17],[0.79,0.81]\right) & \left([0.30,0.33],[0.62,0.65]\right) & \left([0.18,0.19],[0.79,0.80]\right) & \left([0.20,0.25],[0.75,0.75]\right) & \left([0.04,0.06],[0.90,0.93]\right)\\ \left([0.13,0.14],[0.84,0.85]\right) & \left([0.27,0.30],[0.67,0.68]\right) & \left([0.19,0.20],[0.76,0.79]\right) & \left([0.12,0.12],[0.80,0.85]\right) & \left([0.07,0.10],[0.87,0.90]\right)\\ \left([0.14,0.16],[0.79,0.82]\right) & \left([0.26,0.28],[0.70,0.72]\right) & \left([0.20,0.22],[0.75,0.77]\right) & \left([0.15,0.20],[0.75,0.80]\right) & \left([0.02,0.03],[0.94,0.96]\right)\\ \left([0.17,0.20],[0.74,0.80]\right) & \left([0.29,0.32],[0.64,0.67]\right) & \left([0.20,0.23],[0.74,0.76]\right) & \left([0.09,0.12],[0.80,0.85]\right) & \left([0.03,0.04],[0.93,0.94]\right) \end{array}\right) \end{array}$$
$$R^{3}\left(t_{1}\right)=\begin{array}{cc} & \begin{array}{c} \;c_{1}\;c_{2}\;c_{3}\;c_{4}\;c_{5} \end{array}\\ \begin{array}{c} x_{1}\\ x_{2}\\ x_{3}\\ x_{4} \end{array} & \left(\begin{array}{ccccc} \left([0.19,0.20],[0.76,0.78]\right) & \left([0.19,0.21],[0.75,0.77]\right) & \left([0.23,0.24],[0.74,0.76]\right) & \left([0.19,0.24],[0.76,0.76]\right) & \left([0.06,0.09],[0.85,0.87]\right)\\ \left([0.15,0.17],[0.81,0.82]\right) & \left([0.17,0.20],[0.78,0.79]\right) & \left([0.23,0.25],[0.71,0.74]\right) & \left([0.11,0.11],[0.81,0.85]\right) & \left([0.11,0.15],[0.81,0.85]\right)\\ \left([0.16,0.19],[0.75,0.78]\right) & \left([0.16,0.18],[0.80,0.82]\right) & \left([0.25,0.28],[0.69,0.72]\right) & \left([0.15,0.19],[0.76,0.81]\right) & \left([0.03,0.05],[0.91,0.94]\right)\\ \left([0.21,0.24],[0.69,0.76]\right) & \left([0.19,0.21],[0.77,0.79]\right) & \left([0.24,0.28],[0.68,0.70]\right) & \left([0.09,0.11],[0.81,0.85]\right) & \left([0.05,0.09],[0.89,0.91]\right) \end{array}\right) \end{array}$$
$$R^{1}\left(t_{2}\right)=\begin{array}{cc} & \begin{array}{c} \;c_{1}\;c_{2}\;c_{3}\;c_{4}\;c_{5} \end{array}\\ \begin{array}{c} x_{1}\\ x_{2}\\ x_{3}\\ x_{4} \end{array} & \left(\begin{array}{ccccc} \left([0.15,0.16],[0.82,0.83]\right) & \left([0.25,0.28],[0.71,0.72]\right) & \left([0.24,0.25],[0.72,0.74]\right) & \left([0.29,0.38],[0.62,0.62]\right) & \left([0.06,0.06],[0.90,0.92]\right)\\ \left([0.13,0.14],[0.84,0.85]\right) & \left([0.25,0.26],[0.69,0.72]\right) & \left([0.26,0.28],[0.68,0.71]\right) & \left([0.14,0.14],[0.78,0.83]\right) & \left([0.10,0.13],[0.87,0.87]\right)\\ \left([0.12,0.15],[0.82,0.84]\right) & \left([0.23,0.26],[0.70,0.72]\right) & \left([0.30,0.32],[0.66,0.77]\right) & \left([0.22,0.29],[0.71,0.71]\right) & \left([0.03,0.04],[0.92,0.94]\right)\\ \left([0.15,0.17],[0.78,0.80]\right) & \left([0.24,0.25],[0.69,0.74]\right) & \left([0.28,0.31],[0.63,0.67]\right) & \left([0.14,0.14],[0.77,0.83]\right) & \left([0.04,0.06],[0.90,0.92]\right) \end{array}\right) \end{array}$$
$$R^{2}\left(t_{2}\right)=\begin{array}{cc} & \begin{array}{c} \;c_{1}\;c_{2}\;c_{3}\;c_{4}\;c_{5} \end{array}\\ \begin{array}{c} x_{1}\\ x_{2}\\ x_{3}\\ x_{4} \end{array} & \left(\begin{array}{ccccc} \left([0.15,0.16],[0.82,0.83]\right) & \left([0.30,0.32],[0.66,0.68]\right) & \left([0.21,0.22],[0.76,0.78]\right) & \left([0.28,0.31],[0.63,0.63]\right) & \left([0.05,0.05],[0.91,0.93]\right)\\ \left([0.13,0.14],[0.77,0.79]\right) & \left([0.29,0.30],[0.65,0.67]\right) & \left([0.22,0.25],[0.73,0.75]\right) & \left([0.13,0.13],[0.79,0.83]\right) & \left([0.09,0.12],[0.88,0.88]\right)\\ \left([0.12,0.15],[0.82,0.84]\right) & \left([0.27,0.31],[0.65,0.67]\right) & \left([0.26,0.27],[0.70,0.72]\right) & \left([0.21,0.28],[0.72,0.72]\right) & \left([0.03,0.04],[0.93,0.95]\right)\\ \left([0.15,0.17],[0.78,0.80]\right) & \left([0.28,0.30],[0.64,0.69]\right) & \left([0.25,0.27],[0.68,0.71]\right) & \left([0.13,0.13],[0.79,0.83]\right) & \left([0.04,0.06],[0.91,0.93]\right) \end{array}\right) \end{array}$$
$$R^{3}\left(t_{2}\right)=\begin{array}{cc} & \begin{array}{c} \;c_{1}\;c_{2}\;c_{3}\;c_{4}\;c_{5} \end{array}\\ \begin{array}{c} x_{1}\\ x_{2}\\ x_{3}\\ x_{4} \end{array} & \left(\begin{array}{ccccc} \left([0.18,0.19],[0.78,0.80]\right) & \left([0.19,0.21],[0.78,0.79]\right) & \left([0.25,0.27],[0.71,0.73]\right) & \left([0.26,0.35],[0.65,0.65]\right) & \left([0.08,0.08],[0.86,0.89]\right)\\ \left([0.15,0.17],[0.81,0.83]\right) & \left([0.18,0.19],[0.73,0.79]\right) & \left([0.27,0.30],[0.67,0.69]\right) & \left([0.12,0.12],[0.80,0.84]\right) & \left([0.14,0.18],[0.82,0.82]\right)\\ \left([0.14,0.18],[0.78,0.81]\right) & \left([0.17,0.20],[0.78,0.79]\right) & \left([0.32,0.33],[0.64,0.66]\right) & \left([0.20,0.26],[0.74,0.74]\right) & \left([0.04,0.06],[0.89,0.92]\right)\\ \left([0.18,0.20],[0.74,0.77]\right) & \left([0.18,0.19],[0.76,0.80]\right) & \left([0.30,0.33],[0.61,0.65]\right) & \left([0.12,0.12],[0.80,0.84]\right) & \left([0.06,0.08],[0.86,0.89]\right) \end{array}\right) \end{array}$$
$$R^{1}\left(t_{3}\right)=\begin{array}{cc} & \begin{array}{c} \;c_{1}\;c_{2}\;c_{3}\;c_{4}\;c_{5} \end{array}\\ \begin{array}{c} x_{1}\\ x_{2}\\ x_{3}\\ x_{4} \end{array} & \left(\begin{array}{ccccc} \left([0.14,0.16],[0.83,0.84]\right) & \left([0.29,0.32],[0.63,0.67]\right) & \left([0.26,0.29],[0.68,0.71]\right) & \left([0.41,0.41],[0.59,0.59]\right) & \left([0.08,0.09],[0.86,0.90]\right)\\ \left([0.11,0.13],[0.84,0.86]\right) & \left([0.30,0.31],[0.66,0.68]\right) & \left([0.26,0.27],[0.70,0.72]\right) & \left([0.11,0.15],[0.76,0.81]\right) & \left([0.14,0.19],[0.81,0.81]\right)\\ \left([0.12,0.14],[0.85,0.86]\right) & \left([0.31,0.33],[0.63,0.66]\right) & \left([0.25,0.29],[0.66,0.70]\right) & \left([0.31,0.41],[0.59,0.59]\right) & \left([0.05,0.06],[0.90,0.92]\right)\\ \left([0.15,0.17],[0.81,0.83]\right) & \left([0.30,0.32],[0.65,0.67]\right) & \left([0.28,0.30],[0.65,0.69]\right) & \left([0.19,0.24],[0.69,0.76]\right) & \left([0.06,0.06],[0.90,0.92]\right) \end{array}\right) \end{array}$$
$$R^{2}\left(t_{3}\right)=\begin{array}{cc} & \begin{array}{c} \;c_{1}\;c_{2}\;c_{3}\;c_{4}\;c_{5} \end{array}\\ \begin{array}{c} x_{1}\\ x_{2}\\ x_{3}\\ x_{4} \end{array} & \left(\begin{array}{ccccc} \left([0.14,0.16],[0.83,0.84]\right) & \left([0.33,0.37],[0.58,0.62]\right) & \left([0.23,0.25],[0.72,0.75]\right) & \left([0.39,0.39],[0.61,0.61]\right) & \left([0.07,0.09],[0.88,0.91]\right)\\ \left([0.11,0.13],[0.85,0.86]\right) & \left([0.34,0.36],[0.61,0.63]\right) & \left([0.22,0.23],[0.74,0.76]\right) & \left([0.10,0.14],[0.77,0.82]\right) & \left([0.12,0.17],[0.83,0.83]\right)\\ \left([0.12,0.14],[0.85,0.86]\right) & \left([0.36,0.38],[0.58,0.61]\right) & \left([0.17,0.25],[0.71,0.74]\right) & \left([0.29,0.39],[0.61,0.61]\right) & \left([0.04,0.05],[0.91,0.93]\right)\\ \left([0.15,0.17],[0.82,0.83]\right) & \left([0.34,0.37],[0.60,0.62]\right) & \left([0.23,0.26],[0.70,0.73]\right) & \left([0.18,0.23],[0.71,0.77]\right) & \left([0.05,0.05],[0.91,0.93]\right) \end{array}\right) \end{array}$$
$$R^{3}\left(t_{3}\right)=\begin{array}{cc} & \begin{array}{c} \;c_{1}\;c_{2}\;c_{3}\;c_{4}\;c_{5} \end{array}\\ \begin{array}{c} x_{1}\\ x_{2}\\ x_{3}\\ x_{4} \end{array} & \left(\begin{array}{ccccc} \left([0.17,0.19],[0.79,0.80]\right) & \left([0.22,0.24],[0.72,0.75]\right) & \left([0.28,0.31],[0.66,0.69]\right) & \left([0.38,0.38],[0.62,0.62]\right) & \left([0.11,0.14],[0.81,0.86]\right)\\ \left([0.14,0.15],[0.81,0.84]\right) & \left([0.22,0.24],[0.74,0.75]\right) & \left([0.28,0.29],[0.68,0.70]\right) & \left([0.10,0.13],[0.78,0.83]\right) & \left([0.19,0.26],[0.74,0.74]\right)\\ \left([0.15,0.16],[0.82,0.83]\right) & \left([0.24,0.25],[0.72,0.74]\right) & \left([0.27,0.31],[0.64,0.68]\right) & \left([0.29,0.38],[0.62,0.62]\right) & \left([0.06,0.09],[0.86,0.89]\right)\\ \left([0.18,0.20],[0.78,0.79]\right) & \left([0.22,0.25],[0.73,0.75]\right) & \left([0.30,0.32],[0.63,0.67]\right) & \left([0.17,0.22],[0.71,0.78]\right) & \left([0.09,0.09],[0.86,0.89]\right) \end{array}\right) \end{array}$$


Step 4: Use $R^{s}(t_{k})$ and the IVIFWAA operator to aggregate the assessment results of each solution for each attribute, and calculate the WIVIF decision matrix $R(t_{k})$ of three experts at $t_{1}-t_{3}$ using Equations (20)–(23).
$$R\left(t_{1}\right)=\begin{array}{cc} & \begin{array}{c} \;D_{1}\;D_{2}\;D_{3} \end{array}\\ \begin{array}{c} x_{1}\\ x_{2}\\ x_{3}\\ x_{4} \end{array} & \left(\begin{array}{ccc} \left([0.2015,0.2250],[0.7409,0.7610]\right) & \left([0.2150,0.2413],[0.7237,0.7440]\right) & \left([0.1843,0.2057],[0.7642,0.7807]\right)\\ \left([0.1732,0.1988],[0.7697,0.7960]\right) & \left([0.1865,0.2049],[0.7573,0.7783]\right) & \left([0.1610,0.1827],[0.7806,0.8039]\right)\\ \left([0.1795,0.2045],[0.7579,0.7874]\right) & \left([0.1879,0.2126],[0.7536,0.7788]\right) & \left([0.1650,0.1939],[0.7649,0.7966]\right)\\ \left([0.1824,0.2110],[0.7407,0.7765]\right) & \left([0.1980,0.2274],[0.7243,0.7614]\right) & \left([0.1757,0.2059],[0.7442,0.7832]\right) \end{array}\right) \end{array}$$
$$R\left(t_{2}\right)=\begin{array}{cc} & \begin{array}{c} \;D_{1}\;D_{2}\;D_{3} \end{array}\\ \begin{array}{c} x_{1}\\ x_{2}\\ x_{3}\\ x_{4} \end{array} & \left(\begin{array}{ccc} \left([0.2323,0.2603],[0.7248,0.7348]\right) & \left([0.2349,0.2531],[0.7159,0.7292]\right) & \left([0.2045,0.2372],[0.7434,0.7578]\right)\\ \left([0.1780,0.2071],[0.7485,0.7768]\right) & \left([0.2022,0.2158],[0.7299,0.7523]\right) & \left([0.1804,0.2006],[0.7568,0.7862]\right)\\ \left([0.2090,0.2415],[0.7313,0.7582]\right) & \left([0.2117,0.2173],[0.7229,0.7457]\right) & \left([0.1956,0.2272],[0.7463,0.7639]\right)\\ \left([0.1919,0.2130],[0.7241,0.7660]\right) & \left([0.2048,0.2218],[0.6115,0.7573]\right) & \left([0.1885,0.2067],[0.7310,0.7681]\right) \end{array}\right) \end{array}$$
$$R\left(t_{3}\right)=\begin{array}{cc} & \begin{array}{c} \;D_{1}\;D_{2}\;D_{3} \end{array}\\ \begin{array}{c} x_{1}\\ x_{2}\\ x_{3}\\ x_{4} \end{array} & \left(\begin{array}{ccc} \left([0.2780,0.2960],[0.6783,0.7003]\right) & \left([0.2825,0.3050],[0.6664,0.6907]\right) & \left([0.2514,0.2692],[0.7072,0.7271]\right)\\ \left([0.2062,0.2266],[0.7335,0.7583]\right) & \left([0.2196,0.2436],[0.7185,0.7418]\right) & \left([0.1906,0.2145],[0.7488,0.7717]\right)\\ \left([0.2473,0.2949],[0.6760,0.6989]\right) & \left([0.2557,0.3049],[0.6689,0.6886]\right) & \left([0.2196,0.2608],[0.7119,0.7309]\right)\\ \left([0.2272,0.2525],[0.7042,0.7405]\right) & \left([0.2373,0.2686],[0.6956,0.7241]\right) & \left([0.2062,0.2324],[0.7255,0.7606]\right) \end{array}\right) \end{array}$$


Step 5: Use the IVIFWAA and IVIFWGA operators to aggregate the assessment results of the three experts to obtain the target interval-valued intuitionistic fuzzy value at a single time from $t_{1}-t_{3}$, as shown in Table 5 and Table 6.

Step 6: Using Equation (18), take φ=1.5 and obtain the time series weight $\eta_{k}=[0.2000,0.2667,0.5333]$. Further, using the new interval-valued intuitionistic fuzzy budget rule (Definition 7), dynamic multi-time fusion WIVIF decision matrices $H_{\mathrm{IVIFWAA}}$ and $H_{\mathrm{IVIFWGA}}$ based on the IVIFWAA and IVIFWGA operators are constructed.



$$H_{\mathrm{IVIFWAA}}=\begin{array}{cc} & \begin{array}{c} \;t_{1}\;t_{2}\;t_{3} \end{array}\\ \begin{array}{c} x_{1}\\ x_{2}\\ x_{3}\\ x_{4} \end{array} & \left(\begin{array}{ccc} \left([0.0478,0.0547],[0.9351,0.9410]\right) & \left([0.0716,0.0818],[0.9093,0.9144]\right) & \left([0.1655,0.1792],[0.8018,0.8179]\right)\\ \left([0.0404,0.0464],[0.9433,0.9502]\right) & \left([0.0579,0.0655],[0.9162,0.9207]\right) & \left([0.1214,0.1365],[0.8375,0.8538]\right)\\ \left([0.0414,0.0487],[0.9402,0.9488]\right) & \left([0.0646,0.0733],[0.9115,0.9266]\right) & \left([0.1451,0.1771],[0.8084,0.8179]\right)\\ \left([0.0436,0.0519],[0.9331,0.9446]\right) & \left([0.0608,0.0677],[0.8901,0.9238]\right) & \left([0.1334,0.1521],[0.8197,0.8430]\right) \end{array}\right) \end{array}$$


$$H_{\mathrm{IVIFWGA}}=\begin{array}{cc} & \begin{array}{c} \;t_{1}\;t_{2}\;t_{3} \end{array}\\ \begin{array}{c} x_{1}\\ x_{2}\\ x_{3}\\ x_{4} \end{array} & \left(\begin{array}{ccc} \left([0.0475,0.0544],[0.9356,0.9444]\right) & \left([0.0712,0.0816],[0.9096,0.9148]\right) & \left([0.1648,0.1784],[0.8029,0.8189]\right)\\ \left([0.0413,0.0463],[0.9435,0.9505]\right) & \left([0.0577,0.0653],[0.9165,0.9271]\right) & \left([0.1209,0.1359],[0.8382,0.8543]\right)\\ \left([0.0403,0.0486],[0.9403,0.9489]\right) & \left([0.0644,0.0730],[0.9117,0.9207]\right) & \left([0.1440,0.1757],[0.8042,0.8190]\right)\\ \left([0.0434,0.0517],[0.9332,0.9448]\right) & \left([0.0607,0.0676],[0.8960,0.9239]\right) & \left([0.1326,0.1513],[0.8203,0.8438]\right) \end{array}\right) \end{array}$$



Step 7: The positive ideal solutions and negative ideal solutions of $H_{\mathrm{IVIFWAA}}$ and $H_{\mathrm{IVIFWGA}}$ are shown in Table 7. Using Equations (36)–(38), the positive and negative ideal solutions $H_{\mathrm{IVIFWAA}}$, $H_{\mathrm{IVIFWGA}}$, distances $d_{i}^{+}$, $d_{i}^{-}$ of each target $x_{i}$, and the target threat degree $\zeta_{i}$ are calculated, respectively. The calculation results are shown in Table 8.

Specifically, according to Equation (36), the positive ideal distance between target $x_{1}$ and $H_{\mathrm{IVIFWAA}}$ is calculated as follows:



$$\begin{array}{cc} d_{i}^{+} & =\sqrt{\sum_{k=1}^{3}\left[\frac{1}{4}\left[{\left(\mu_{ik}^{L}-\mu_{k}^{L+}\right)}^{2}+{\left(\mu_{ik}^{U}-\mu_{k}^{U+}\right)}^{2}+{\left(v_{ik}^{L}-v_{k}^{L+}\right)}^{2}+{\left(v_{ik}^{U}-v_{k}^{U+}\right)}^{2}+{\left(\mu_{ik}^{L}+v_{ik}^{L}-\mu_{k}^{L+}-v_{k}^{L+}\right)}^{2}+{\left(\mu_{ik}^{U}+v_{ik}^{U}-\mu_{k}^{U+}-v_{k}^{U+}\right)}^{2}\right]\right]}\\ & =\sqrt{\begin{array}{cc} \; & [\frac{1}{4}{\left(0.0478-0.0478\right)}^{2}+{\left(0.0547-0.0547\right)}^{2}+{\left(0.9351-0.9331\right)}^{2}+{\left(0.9410-0.9410\right)}^{2}+{\left(0.0478+0.9351-0.0478-0.9331\right)}^{2}\\ \; & +{\left(0.0547+0.9410-0.0547-0.9410\right)}^{2}]+[\frac{1}{4}{\left(0.0716-0.0716\right)}^{2}+{\left(0.0818-0.0818\right)}^{2}+{\left(0.9093-0.8901\right)}^{2}+{\left(0.9144-0.9144\right)}^{2}\\ \; & +{\left(0.0716+0.9093-0.0716-0.8901\right)}^{2}+{\left(0.0818+0.9144-0.0818-0.9144\right)}^{2}]+0 \end{array}}\\ & =0.0136 \end{array}$$



It can be seen from Table 8 that the target multi-time fusion threat ordering obtained by the IVIFWAA operator and the IVIFWGA operator is Target 1 > Target 3 > Target 4 > Target 2, which verifies the feasibility of this algorithm.

### 6.2. Algorithm Superiority Test and Analysis

In the missile defense dynamic fusion target threat assessment, time is an important factor that affects the decision result. Table 9 and Figure 3 show the target threat degree and ordering results of the single times $t_{1}-t_{3}$ and dynamic multi-time fusion. We see that the target threat ordering results at different moments are roughly the same, and the targets with the largest and smallest threat levels are consistent. However, even if the target threat degree order is the same, the threat degree of each target is different at different times, and Target 4 and Target 3 are more sensitive to changes in time. At $t_{1}$ and $t_{2}$, the threat of Target 4 is higher than that of Target 3 and the opposite is true at t3. It can be seen that the static threat assessment method at a single moment cannot reflect the timing of the target and the dynamic changes of the battlefield. The method proposed in this paper not only considers the tendency of Target 4 to decrease in speed, distance, and RCS threat degree during the entire assessment process, but also considers the threat of a sudden increase in speed, RCS, interference intensity, and defense capability of Target 3 at $t_{3}$. Therefore, we can obtain more reliable threat assessment results.

Additionally, in the ever-changing missile defense combat of the battlefield, the accuracy of the dynamic fusion target threat assessment method mainly depends on the difference between target threat degrees in the same assessment method. The more obvious the difference, the better one is able to select the optimal solution, that is, the stronger the superiority of the method. Therefore, the superior degree (SD) of target i over target j can be defined as:$$SD_{ij}(\%)=(\frac{\zeta_{i}-\zeta_{j}}{\zeta_{i}})\times 100\%$$
where $\zeta_{i}$ and $\zeta_{j}$ are the threat degree $\left\{i=1,2,\right.\left.\cdots ,m;j=1,2,\cdots ,m;i\neq j\right\}$ of different targets.

In order to verify the effectiveness of the method proposed in this paper, the method is compared with UDIFWA operator, DINFWAA operator, CIIFA operator, IVIFPWA operator, and D-S-P operator proposed in [30,31,32,33,34]. The target threat degree ordering results and superiority of different methods are shown in Figure 4 and Table 10. The target multi-time fusion threat ordering result obtained by using the IVIFWAA operator and the IVIFWGA operator in this paper is the same as in references [30,31,32,33], which shows the effectiveness of the method proposed in this paper. The ordering results are slightly different from those in reference [34] but the final decision of the optimal solution is the same.

Note that in the method proposed in this article, the superiority gap between the targets is the largest. When comparing the superiority of Target 1 and Target 3, the superiority of the algorithm in this paper (IVIFWGA) is 1.67, 1.68, 1.92, and 1.69 times that of the methods proposed in [30,31,32,33], respectively. In the superiority comparison of Target 3 and Target 4, the superiority of the algorithm in this paper (IVIFWGA) is 1.53, 1.61, 1.92, and 2.41 times that of the methods proposed in [30,31,32,33], respectively. In the superiority comparison of Target 4 and Target 2, the superiority of the algorithm in this paper (IVIFWGA) is 1.41, 1.74, 1.53, and 1.71 times that of the methods proposed in [30,31,32,33], respectively. The greater the superiority gap, the easier it is for the commander to make decisions. This demonstrates that the method in this paper, by integrating the subjective and objective weights of each attribute of the target and the weight of the time series, considers the degree of change in the relative difference of each attribute and uses the IVIFWAA/IVIFWGA operator to combine decision-making information of multiple target attributes, multiple moments, and multiple experts. This effectively avoids the problem of decision-making errors due to unclear superiority under the influence of subjective factors.

## 7. Conclusions

This paper combines interval-valued intuitionistic fuzzy set theory with dynamic multi-attribute group decision-making theory, the dynamic fusion target threat assessment method for missile defense is proposed. By comparison with static threat assessment and existing dynamic threat assessment methods, the feasibility and superiority of the method in this paper is verified. The main contributions of the proposed model are as follows: (1) The new interval-valued intuitionistic fuzzy weighted average operator based on the definition of new interval-valued intuitionistic fuzzy operation rules are proposed. (2) An integrated weight model of target attributes based on IVIFECF is proposed to solve the problem of inconsistency with intuitive facts when the deviation between membership and non-membership are equal. (3) Ordering method of interval-valued intuitionistic fuzzy numbers based on improved TOPSIS is proposed to improve the ability to distinguish and differentiate between decision-making results. (4) In order to improve the reliability and accuracy of missile defense target threat assessment, an assessment model based on IVIFEC-IVIFWA-TOPSIS is constructed, and the result of dynamic fusion target threat assessment considering the fusion of multi-target attributes, multiple times, and multi-expert decision information is obtained.

In this paper, the influence of the battlefield situation on target threat assessment has not been considered, and the intelligence level of target threat assessment method needs to be improved. Therefore, future research will focus on the following aspects: (1) Carefully considering the impact of the battlefield situation on target threat in complex battlefield environment. (2) Applying intelligent simulation technologies, such as reinforcement learning and deep learning, to threat assessment. (3) Research on target threat assessment method under incomplete target information.

## Figures and Tables

**Figure 1 entropy-24-01825-f001:**
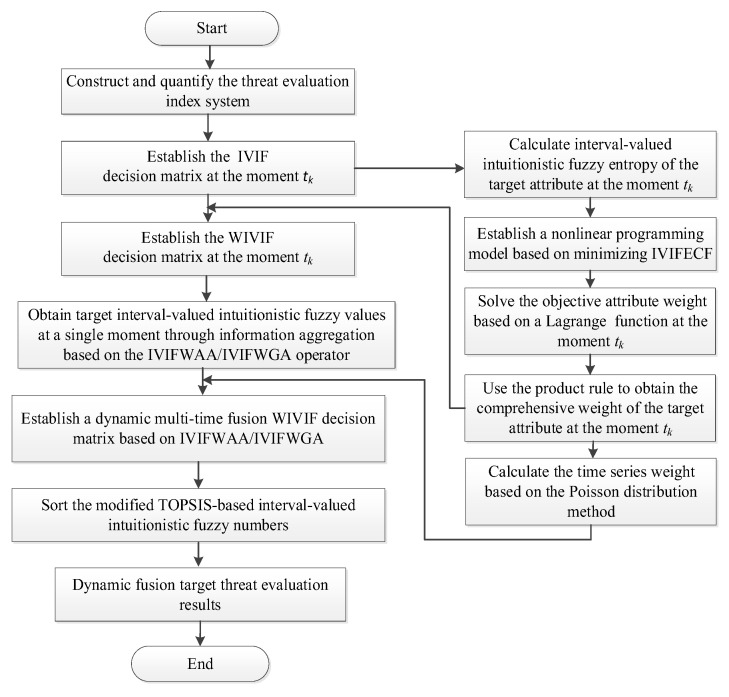
Flow chart of the IVIFECF-IVIFWA-TOPSIS assessment model.

**Figure 2 entropy-24-01825-f002:**
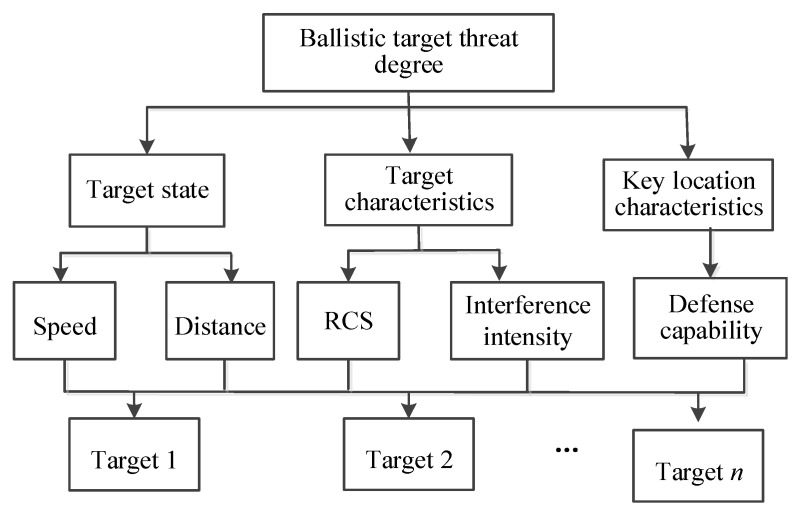
Target threat assessment index system for missile defense combat.

**Figure 3 entropy-24-01825-f003:**
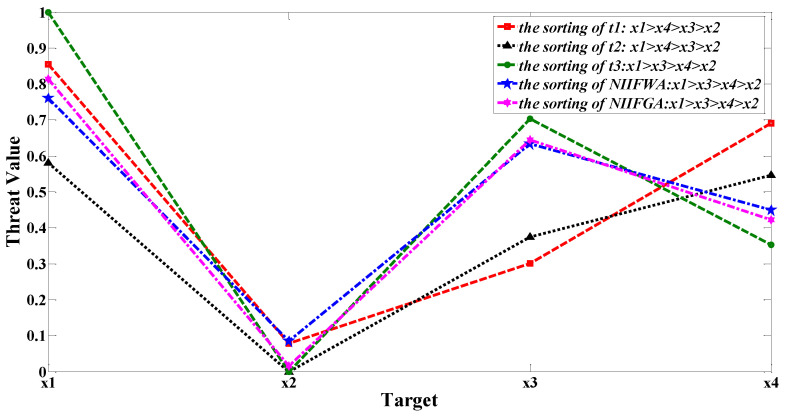
Comparison of target ordering results between time points of $t_{1}-t_{3}$ single time and dynamic multi-time aggregation.

**Figure 4 entropy-24-01825-f004:**
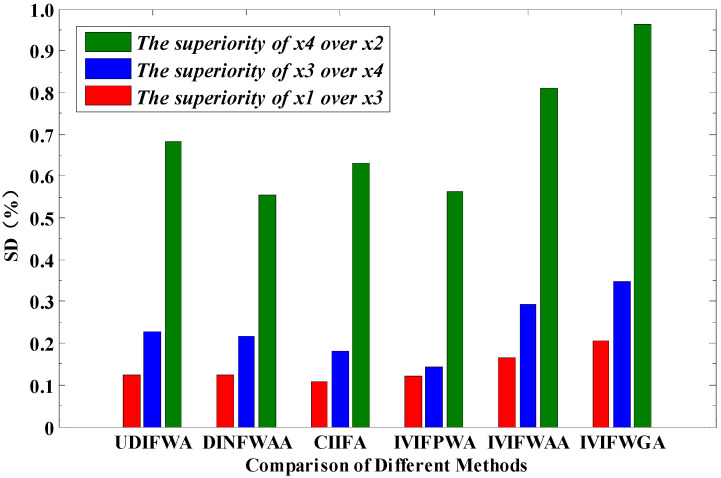
Comparison of superiority of different methods.

**Table 1 entropy-24-01825-t001:** Correspondence between 9-level quantification results and interval-valued intuitionistic fuzzy numbers.

Threat Level	Symbol	Interval-Valued Intuitionistic Fuzzy Numbers
Extremely big	EB	$\left(\left[0.9,0.9\right],[0.1,0.1]\right)$
Very big	VB	$\left(\left[0.8,0.9\right],[0.1,0.1]\right)$
Big	B	$\left(\left[0.7,0.8\right],\left[0.2,0.2\right]\right)$
Medium big	MB	$\left(\left[0.6,0.7\right],\left[0.2,0.3\right]\right)$
Medium	M	$\left(\left[0.5,0.5\right],\left[0.3,0.4\right]\right)$
Medium small	MS	$\left(\left[0.4,0.5\right],\left[0.3,0.4\right]\right)$
Small	S	$\left(\left[0.3,0.4\right],\left[0.4,0.5\right]\right)$
Very small	VS	$\left(\left[0.2,0.3\right],\left[0.5,0.6\right]\right)$
Extremely small	ES	$\left(\left[0.1,0.2\right],\left[0.6,0.7\right]\right)$

**Table 2 entropy-24-01825-t002:** Calculation results of Example 1 with different entropy power methods.

Method	Entropy	Comparison Results
Entropy of IVIFSs	$E_{1}\left(\tilde{A}\right)=0.800,E_{1}\left(\tilde{B}\right)=0.800$	$E\left(\tilde{A}\right)=E\left(\tilde{B}\right)$
Entropy measures of IVIFSs	$E_{2}\left(\tilde{A}\right)=0.958,E_{2}\left(\tilde{B}\right)=0.958$	$E\left(\tilde{A}\right)=E\left(\tilde{B}\right)$
This paper’s method	$E_{2}\left(\tilde{A}\right)=0.966,E_{2}\left(\tilde{B}\right)=0.951$	$E\left(\tilde{A}\right)>E\left(\tilde{B}\right)$

**Table 3 entropy-24-01825-t003:** Target attribute data at $t_{1}-t_{3}$.

Time	Target	Speed	Distance	RCS	Interference	Defense
$t_{1}$	$x_{1}$	([0.52,0.55], [0.37,0.41])	([0.65,0.69], [0.24,0.28])	([0.70,0.72], [0.24,0.27])	([0.7,0.8], [0.2,0.2])	([0.4,0.5], [0.3,0.4])
	$x_{2}$	([0.44,0.49], [0.48,0.50])	([0.61,0.66], [0.30,0.32])	([0.71,0.74], [0.20,0.25])	([0.5,0.5], [0.3,0.4])	[0.6,0.7], [0.2,0.3])
	$x_{3}$	([0.47,0.53], [0.36,0.42])	([0.59,0.63], [0.34,0.37])	([0.74,0.78], [0.18,0.21])	([0.6,0.7], [0.2,0.3])	[0.2,0.3], [0.5,0.6])
	$x_{4}$	([0.56,0.62], [0.27,0.38])	([0.64,0.68], [0.27,0.31])	([0.73,0.79], [0.16,0.19])	([0.4,0.5], [0.3,0.4])	[0.3,0.4], [0.4,0.5])
$t_{2}$	$x_{1}$	([0.53,0.56], [0.39,0.43])	([0.66,0.70], [0.28,0.30])	([0.72,0.74], [0.22,0.25])	([0.8,0.9], [0.1,0.1])	([0.5,0.5], [0.3,0.4])
	$x_{2}$	([0.47,0.51], [0.45,0.48])	([0.65,0.67], [0.26,0.29])	([0.75,0.79], [0.17,0.20])	([0.5,0.5], [0.3,0.4])	([0.7,0.8], [0.2,0.2])
	$x_{3}$	([0.45,0.54], [0.39,0.44])	([0.62,0.68], [0.27,0.29])	([0.81,0.83], [0.14,0.16])	([0.7,0.8], [0.2,0.2])	[0.3,0.4], [0.4,0.5])
	$x_{4}$	([0.53,0.57], [0.31,0.36])	([0.63,0.66], [0.25,0.32])	([0.79,0.82], [0.12,0.15])	([0.5,0.5], [0.3,0.4])	([0.4,0.5], [0.3,0.4]
$t_{3}$	$x_{1}$	([0.55,0.59], [0.38,0.40])	([0.70,0.74], [0.20,0.24])	([0.78,0.81], [0.15,0.18])	([0.9,0.9], [0.1,0.1])	([0.6,0.7], [0.2,0.3])
	$x_{2}$	([0.46,0.50], [0.42,0.47])	([0.71,0.73], [0.23,0.25])	([0.77,0.79], [0.17,0.20])	([0.4,0.5], [0.3,0.4])	([0.8,0.9], [0.1,0.1])
	$x_{3}$	([0.49,0.53], [0.43,0.46])	([0.73,0.76], [0.20,0.23])	([0.76,0.82], [0.13,0.17])	([0.8,0.9], [0.1,0.1])	([0.4,0.5], [0.3,0.4])
	$x_{4}$	([0.57,0.61], [0.35,0.38])	([0.71,0.75], [0.22,0.24])	([0.80,0.83], [0.12,0.16])	([0.6,0.7], [0.2,0.3])	([0.5,0.5], [0.3,0.4])

**Table 4 entropy-24-01825-t004:** Interval-valued intuitionistic fuzzy entropy of target attributes at different moments.

Time	IVIFECF Value				
	Speed	Distance	RCS	Interference Intensity	Defense Capability
$t_{1}$	0.9730	0.8716	0.7037	0.9101	0.9370
$t_{2}$	0.9817	0.8478	0.6092	0.7807	0.9232
$t_{3}$	0.9752	0.7277	0.5776	0.6439	0.8114

**Table 5 entropy-24-01825-t005:** IVIFWAA operator aggregation assessment results of all decision makers.

Target	IVIFWAA Operator Aggregation		
	$t_{1}$	$t_{2}$	$t_{3}$
$x_{1}$	([0.2006,0.2244],[0.7423,0.7614])	([0.2242,0.2504],[0.7277,0.7403])	([0.2711,0.2905],[0.6833,0.7055])
$x_{2}$	([0.1738,0.1957],[0.7690,0.7924])	([0.1872,0.2080],[0.7447,0.7713])	([0.2058,0.2286],[0.7332,0.7569])
$x_{3}$	([0.1777,0.2038],[0.7587,0.7874])	([0.2056,0.2287],[0.7332,0.7557])	([0.2414,0.2875],[0.6923,0.7055])
$x_{4}$	([0.1856,0.2149],[0.7362,0.7734])	([0.1953,0.2140],[0.6834,0.7637])	([0.2240,0.2517],[0.7081,0.7412])

**Table 6 entropy-24-01825-t006:** IVIFWGA operator aggregation assessment results of all decision makers.

Target	IVIFWGA Operator Aggregation		
	$t_{1}$	$t_{2}$	$t_{3}$
$x_{1}$	([0.1996,0.2234],[0.7438,0.7625])	([0.2232,0.2500],[0.7285,0.7412])	([0.2701,0.2893],[0.6848,0.7069])
$x_{2}$	([0.1734,0.1953],[0.7695,0.7933])	([0.1866,0.2075],[0.7454,0.7722])	([0.2050,0.2277],[0.7342,0.7577])
$x_{3}$	([0.1772,0.2036],[0.7590,0.7879])	([0.2051,0.2280],[0.7336,0.7560])	([0.2398,0.2856],[0.6865,0.7070])
$x_{4}$	([0.1850,0.2143],[0.7365,0.7738])	([0.1950,0.2137],[0.6968,0.7640])	([0.2228,0.2506],[0.7089,0.7423])

**Table 7 entropy-24-01825-t007:** Positive and negative ideal solutions of $H_{\mathrm{IVIFWAA}}$ and $H_{\mathrm{IVIFWGA}}$.

$H_{\mathrm{IVIFWAA}}$	Positive	([0.0478,0.0547],[0.9331,0.9410])	([0.0716,0.0818],[0.8901,0.9144])	([0.1655,0.1792],[0.8018,0.8179])
	Negative	([0.0404,0.0464],[0.9433,0.9502])	([0.0579,0.0655],[0.9162,0.9266])	([0.1214,0.1365],[0.8375,0.8538])
$H_{\mathrm{IVIFWGA}}$	Positive	([0.0475,0.0544],[0.9332,0.9444])	([0.0712,0.0816],[0.8960,0.9148])	([0.1648,0.1784],[0.8029,0.8189])
	Negative	([0.0403,0.0463],[0.9435,0.9505])	([0.0577,0.0653],[0.9165,0.9271])	([0.1209,0.1359],[0.8382,0.8543])

**Table 8 entropy-24-01825-t008:** The positive and negative ideal solution distance and threat degree of $H_{NIIFWA}$ and $H_{NIIFGA}$.

Result	*H* _IVIFWAA_				*H* _IVIFWGA_			
	*x* _1_	*x* _2_	*x* _3_	*x* _4_	*x* _1_	*x* _2_	*x* _3_	*x* _4_
$d_{i}^{+}$	0.0136	0.0453	0.0227	0.0297	0.0098	0.0439	0.0192	0.0295
$d_{i}^{-}$	0.0431	0.0042	0.0393	0.0242	0.0427	0.0007	0.0349	0.0215
$\zeta_{i}$	0.7601	0.0848	0.6339	0.4490	0.8133	0.0157	0.6451	0.4216

**Table 9 entropy-24-01825-t009:** Target threat degree and ordering results at different moments.

Time	$x_{1}$	$x_{2}$	$x_{3}$	$x_{4}$	Ordering Results
$t_{1}$	0.8549	0.0787	0.3006	0.6909	1 > 4 > 3 > 2
$t_{2}$	0.5804	0.0000	0.3750	0.5458	1 > 4 > 3 > 2
$t_{3}$	1.0000	0.0000	0.7031	0.3525	1 > 3 > 4 > 2
This paper’s method(based on IVIFWAA)	0.7601	0.0848	0.6339	0.4490	1 > 3 > 4 > 2
This paper’s method(based on IVIFWGA)	0.8133	0.0157	0.6451	0.4216	1 > 3 > 4 > 2

**Table 10 entropy-24-01825-t010:** Target threat ordering and superiority of different methods.

Method	$x_{1}$	$x_{2}$	$x_{3}$	$x_{4}$	Ordering Results	$SD_{ij}(\%)$		
UDIFWA operator	0.8014	0.1729	0.7026	0.5451	1 > 3 > 4 > 2	SD_13_ = 0.1233	SD_34_ = 0.2270	SD_42_ = 0.6828
DINFWAA operator	0.6452	0.1982	0.5659	0.4438	1 > 3 > 4 > 2	SD_13_ = 0.1229	SD_34_ = 0.2158	SD_42_ = 0.5534
CIIFA operator	0.9203	0.2489	0.8215	0.6734	1 > 3 > 4 > 2	SD_13_ = 0.1074	SD_34_ = 0.1803	SD_42_ = 0.6304
IVIFPWA operator	0.7329	0.2406	0.6437	0.5513	1 > 3 > 4 > 2	SD_13_ = 0.1217	SD_34_ = 0.1435	SD_42_ = 0.5636
D-S-P operator	0.9047	0.3138	0.6492	0.8020	1 > 4 > 3 > 2	SD_14_ = 0.1135	SD_43_ = 0.1905	SD_32_ = 0.5166
IVIFWAA operator	0.7601	0.0848	0.6339	0.4490	1 > 3 > 4 > 2	SD_13_ = 0.1660	SD_34_ = 0.2917	SD_42_ = 0.8111
IVIFWGA operator	0.8133	0.0157	0.6451	0.4216	1 > 3 > 4 > 2	SD_13_ = 0.2068	SD_34_ = 0.3465	SD_42_ = 0.9627

## Data Availability

Not applicable.

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
