# Peer review of "A New Model of Interval-Valued Intuitionistic Fuzzy Weighted Operators and Their Application in Dynamic Fusion Target Threat Assessment"

_entropy, 2022, doi:10.3390/e24121825_

Round 1
Reviewer 1 Report
Thank you for inviting me as a reviewer for the paper titled “A new model of interval-valued intuitionistic fuzzy weighted operators and their application in dynamic fusion target threat assessment”. First of all, the aims and scope of the paper match those of Entropy. The manuscript is very interesting and well organized. In this paper, the authors gave a comprehensive model that considers: the subjective weights, objective weights, and the fusion of multi-target attributes, multi-time information, and multi-expert decision information. The proposed model was applied in missile defense target threat assessment.
Although the paper is very well prepared, I think there are minor omissions that need to be corrected:
- Expand the literature analysis. Analyze recent papers, period 2020-2022 (the latest reference the authors refer to is from 2020, and only one; the other references are older). What about new research, such as Pamučar, D., & Janković, A. (2020). The application of the hybrid interval rough weighted Power-Heronian operator in multi-criteria decision making. Operational Research in Engineering Sciences: Theory and Applications, 3(2), 54-73. https://doi.org/10.31181/oresta2003049p; Anusha, G., Ramana, P. V., & Sarkar, R. (2022). Hybridizations of Archimedean copula and generalized MSM operators and their applications in interactive decision-making with q-rung probabilistic dual hesitant fuzzy environment. Decision Making: Applications in Management and Engineering. https://doi.org/10.31181/dmame0329102022a; Pamucar, D. (2020). Normalized weighted geometric Dombi Bonferroni mean operator with interval grey numbers: Application in multicriteria decision making. Reports in Mechanical Engineering, 1(1), 44-52. https://doi.org/10.31181/rme200101044p.
- It Will be useful to add some examples of calculation in steps 1-7 in section 6.1. This will bring the paper closer to a larger number of readers.
- In table 10 and figure 4, authors use: “references [30]”, “references [31]”, … Instead word “reference” need to use the name of the operator. Table 2 is a similar situation.
- The conclusion - Clearly state your unique research contributions in the conclusion section. Provide some future directions. Limitations of your study should be added, also.
Reviewer 2 Report
Dear authors,
it is a pleasure for me to review your paper. Thanks for your contribution into the interval-valued intuitionistic fuzzy weighted operators and their application in dynamic fusion target threat assessment. I find it quite interesting and useful for the audience.
Though I recommend you to pay your attention on some grammar mistakes or misprints like in line 367,375,384 and correct them.
I would also extend the list of references a bit.
In general, your paper is worth publishing.
many thanks and best regards
Reviewer 3 Report
The paper is well-written, but the contribution is not significant. My comments are as follows:
- - Please highlight the contributions of the current study in the introduction section.
- - Please cite recent MCDM methods and their fuzzy version, which are also related to this manuscript. For example, I suggest you see the Fuzzy Ordinal Priority Approach.
- - A comparative analysis can be provided.
- - The conclusion needs significant improvement. Especially mention the limitations of the current study clearly.
Round 2
Reviewer 1 Report
All the reviewers' comments have been addressed carefully and sufficiently. The revisions are rational from my point of view. I think the current version of the paper can be accepted.
Reviewer 3 Report
The authors did not apply my first comment in the previous review. I will give them another chance to revise the manuscript. Moreover, the language of the manuscript should be improved.
